



# Case study of a humidity layer above Arctic stratocumulus using balloon-borne turbulence and radiation measurements and large eddy simulations

Ulrike Egerer[1], André Ehrlich[2], Matthias Gottschalk[2], Roel A. J. Neggers[3], Holger Siebert[1], and Manfred Wendisch[2]

[1]Leibniz Institute for Tropospheric Research, Permoserstr. 15, 04318 Leipzig, Germany
[2]Leipzig Institute for Meteorology, University of Leipzig, Stephanstr. 3, 04103 Leipzig, Germany
[3]Institute for Geophysics and Meteorology, University of Cologne, Pohligstr. 3, 50969 Cologne, Germany

**Correspondence:** Ulrike Egerer (egerer@tropos.de)

**Abstract.** Specific humidity inversions occur frequently in the Arctic. The formation of these inversions is often associated with large scale advection of humid air. However, small-scale boundary layer processes interacting with the humidity inversions are not fully understood yet. In this study, we analyze a three-day period of a persistent layer of increased specific humidity above a stratocumulus cloud observed during an Arctic field campaign in June 2017. The tethered balloon system BELUGA

5  (Balloon-bornE moduLar Utility for profilinG the lower Atmosphere) recorded high-resolution vertical profile measurements of turbulence and radiation in the atmospheric boundary layer. We find that the humidity inversion and the cloud layer are coupled by eddy dissipation, extending above the cloud boundary and linking both layers through turbulent mixing. One case reveals a strong negative virtual sensible heat flux at cloud top (eddy covariance estimate of -15 $\mathrm{W\,m^{-2}}$), indicating entrainment of humid air from above into the cloud layer. Large Eddy Simulations (LES) based on field campaign data are conducted to

10  supplement the flux measurements. Independent experiments for two days confirm the observed entrainment of humid air, reproducing the observed negative turbulent fluxes of heat and moisture at cloud top. The LES realizations suggest that in the presence of a humidity layer the cloud layer remains thicker and the inversion height is slightly raised, reproducing results from previous idealized LES studies. While this acts to prevent cloud collapse, it remains unclear how the additional moisture is processed in the cloud and how exactly it contributes to the longevity of Arctic cloud layers.

## 1 Introduction

The Arctic atmospheric boundary layer (ABL) exhibits numerous peculiarities compared to lower latitudes, such as persistent mixed-phase clouds, multiple cloud layers decoupled from the surface and ubiquitous vertical temperature inversions close to the ground. Local ABL and cloud processes are complex and not completely understood, but they are considered an important component to explain the rapid warming of the Arctic region (Wendisch et al., 2019). One of the special features frequently

20  observed in the Arctic are specific humidity inversions (SHIs), although specific humidity is generally expected to decrease





with height (Nicholls and Leighton, 1986; Wood, 2012). The frequency of occurrence of low level SHIs in summer is estimated to be in the range of 70–90 % over the Arctic ocean (Naakka et al., 2018).

Arctic SHIs have been observed during past field campaigns (Sedlar et al., 2012; Pleavin, 2013), e.g. the Surface Heat Budget of the Arctic Ocean (SHEBA;  Uttal et al., 2002) in 1997/98, or the Arctic Summer Cloud Ocean Study (ASCOS; Tjernström et al., 2014) in 2008. Furthermore, a number of studies about the climatology of SHIs have been published (e.g. Naakka et al., 2018; Brunke et al., 2015). Over the Arctic ocean, SHIs occur most frequently and are strongest in summer. In the lower troposphere, they often occur in conjunction with temperature inversions and high relative humidity, but they also depend on the surface energy budget (Naakka et al., 2018). Formation processes and interactions of SHIs with clouds have been investigated in Large Eddy Simulations (LES). For example, Solomon et al. (2014) showed that a specific humidity layer becomes important as a moisture source for the cloud, when moisture supply from the surface is limited. Pleavin (2013) studied how the SHIs support the mixed phase clouds to extend into the temperature and humidity inversion.

Mostly, the formation of the summertime SHIs is attributed to large-scale advection of humid air masses. In the Arctic, especially over sea ice, moisture advection is the critical factor for cloud formation and development (Sotiropoulou et al., 2018). SHIs form when warm, moist continental air is advected over the cold sea surface and moisture is removed through condensation and precipitation from the lowest ABL part. This and further simplified formation processes are discussed in Naakka et al. (2018).

SHIs can contribute to the longevity of Arctic mixed-phase clouds (Morrison et al., 2012; Sedlar and Tjernström, 2009), which influence the near-surface radiation heat budget (Intrieri et al., 2002). When an SHI is located above a cloud, it can provide moisture for the cloud due to cloud top entrainment. In contrast, in the typical marine sub-tropical or mid-latitude cloud topped ABL, dry air from above is entrained into the cloud (Albrecht et al., 1985; Nicholls and Leighton, 1986; Katzwinkel et al., 2012). Despite their importance for the near-surface energy budget, SHIs are not well represented in global atmospheric models, where the SHI strength is typically underestimated (Naakka et al., 2018), or the SHIs are not reproduced (Sotiropoulou et al., 2016).

Previous studies about SHIs are based on radiosoundings, remote sensing observations, reanalysis data or LES. Local small-scale in situ observations of SHIs are missing to characterize and quantify turbulent and radiation properties. However, vertical moisture transport close to the cloud top is key to understand the importance of SHIs for the cloud lifetime. Therefore, we perform tethered balloon-borne, high-resolution vertical profile measurements of turbulence and radiation recorded within a three-day period during the Physical Feedbacks of Arctic Boundary Layer, Sea Ice, Cloud and Aerosol (PASCAL) campaign (Wendisch et al., 2019), combined with LES for the same period. We focus on a detailed case study with a persistent SHI above a stratocumulus deck. Using the observations and simulations, we investigate the local ABL structure around the SHI and study the turbulent transport between the SHI and the cloud layer.





## 2 Observational

### 2.1 The PASCAL expedition

The observations analyzed in this study were performed during PASCAL (Wendisch et al., 2019), which took place in the
sea-ice covered area north of Svalbard in summer 2017. The RV *Polarstern* (Knust, 2017) carried a suite of remote sensing
and in situ instrumentation. Additionally, an ice floe camp was erected in the vicinity of the ship (Macke and Flores, 2018).
Knudsen et al. (2018) describe the ice floe period as climatologically warm with warm and moist maritime air masses advected
from the South and East. The present study is based on measurements with instruments carried by the tethered balloon system
BELUGA (Balloon-bornE moduLar Utility for profilinG the lower Atmosphere;  Egerer et al., 2019b). BELUGA was launched
from the ice floe at around $82°$ N, $10°$ E in the period of 5–14 June 2017. The balloon measurements are complemented by
radiosoundings launched every six hours (Schmithüsen, 2017) and ship-based remote sensing data from radar and lidar, which
are processed with the Cloudnet algorithm (Griesche et al., 2019a; Griesche et al., 2019).

### 2.2 BELUGA setup

The BELUGA system consists of a $90 \ \mathrm{m}^3$ helium-filled tethered balloon with a modular set-up of different instrument pack-
ages to explore the ABL between the surface and about 1500 m altitude. BELUGA can operate under cloudy and light icing
conditions in the Arctic. Fixed to the balloon tether, a fast (50 Hz resolution) ultrasonic anemometer supported by an inertial
navigation system measures the wind velocity vector in an Earth-fixed coordinate system together with the virtual air tem-
perature. Furthermore, barometric pressure, relative humidity, and the static temperature are measured with lower resolution.
A second instrument payload is simultaneously fixed to the tether, measuring broadband terrestrial and solar net irradiances.
Technical details on BELUGA, its instrumentation and operation during PASCAL, as well as data processing methods are
given in Egerer et al. (2019b).

### 2.3 Humidity measurements under cloudy conditions

Humidity and temperature measurements are challenging under cloudy and cold conditions. Specific humidity $q$ is derived
from measurements of air temperature $T$ and relative humidity RH. Those parameters are obtained by regular radiosoundings
(Vaisala RS92-SGP) and the BELUGA system with similar capacitive sensors ascending through the cloud layer, and therefore
suffer from similar limitations. The main challenge is wetting of the sensors during the cloud penetration. A water film on the
sensor might increase the response time (as the water needs to evaporate first) in the sub-saturated air above clouds during an
ascent. This might significantly influence the air temperature and humidity measurements. A detailed discussion on wetting
and icing problems of radiosondes is provided by Jensen et al. (2016), showing that wet-bulbing is an issue for the radiosonde
type used during PASCAL. Artefacts of this process can thus be present in the humidity and temperature profiles as sampled
by radiosondes. This is particularly relevant for studies of humidity inversions in the Arctic, most of which have made use of
such radiosoundings. Because radiosondes penetrate the cloud layer from below, the impact of wet-bulbing is most pronounced





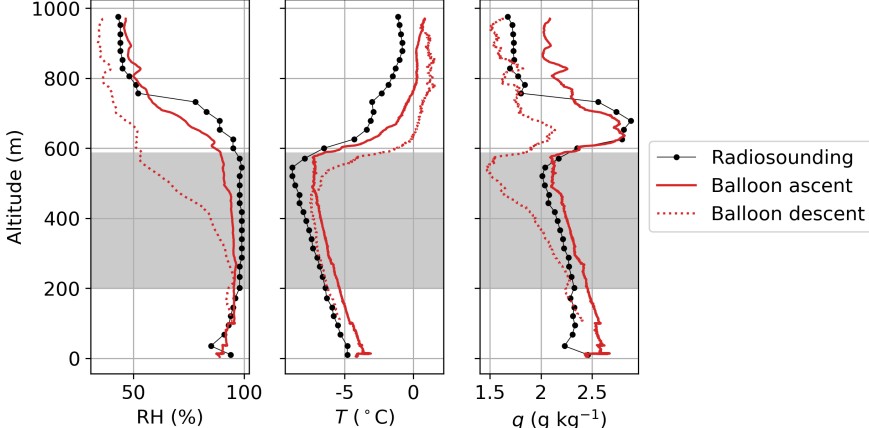

**Figure 1.** Vertical profile of relative humidity RH, Temperature $T$ and specific humidity $q$ measured by a radiosonde and BELUGA on 7 June 2017. The radiosounding was launched at 11:00 UTC, the balloon flew a continuous ascent and descent from 9:30 to 10:30 UTC. The cloud extent (from Cloudnet data) is shown as shaded area.

when it leaves the cloud layer - which is unfortunately exactly the height range where the humidity anomaly is situated. This puts some doubt on radiosonde recordings of humidity inversions.

A simple and convincing test of the influence of possible wet-bulbing on the observations of SHIs is a measurement in the opposite direction, that is a descent from the free troposphere through the area with increased specific humidity into the cloud layer. This is not done with regular radiosoundings, but feasible for the BELUGA operation.

    Figure 1 shows vertical profiles of RH, $T$ and $q$ as measured by both platforms (radiosonde and BELUGA) on 7 June 2017. The launch time of the radiosonde and the balloon differs by around 1.5 hours, which means that discrepancies in the measure-

ments can also be attributed to changes in environmental conditions. Qualitatively, radiosounding and BELUGA measurements show a similar vertical structure. Both observations show a layer of increased specific humidity, hereafter referred to as humidity layer, between about 600 m and 750 m altitude. The increased specific humidity emerges from relative humidity remaining close to saturation within the temperature inversion, before decreasing to the free troposphere level well above the inversion base. The case on 7 June is selected as an illustrative example, because it is the only BELUGA flight of the study period with a

combination of a continuous ascent and subsequent descent. The descent also shows an increase of specific humidity, although the RH profile suggests reduced cloud height and thickness. The wet-bulbing effect cannot be quantified at this point, but despite the difference between ascent and descent the main vertical structure is similar. Hence, we conclude that the observation of the increased specific humidity is real and not the result of a measurement artefact.

    The vertical cloud extent plays an important role for analyzing the vertical structure of specific humidity. The cloud bound-

aries in Fig. 1 are estimated from Cloudnet data (Griesche et al., 2019) for the time of the balloon ascent and reflect the water

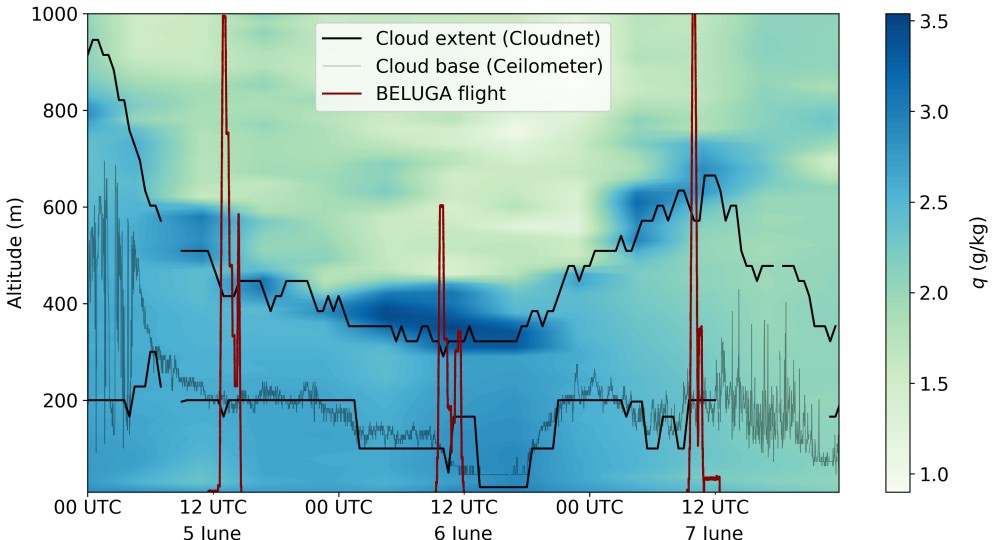

**Figure 2.** Temporal development of the specific humidity vertical profile observed by radiosondes. The period 5 to 7 June 2017 exhibits a distinct layer of increased humidity above a low, single-layer stratocumulus. The cloud extent derived from Cloudnet data is depicted as black lines, the cloud base height derived from the *Polarstern* ceilometer data (Schmithüsen, 2018) is indicated as a grey line. The red lines represent the BELUGA flight profiles.

and ice cloud. Unfortunately, the BELUGA instrumentation does not allow for in situ measurements of cloud liquid water to estimate the cloud boundaries more precisely.

## 3 Case study

### 3.1 Three-day period of a persistent humidity layer

A persistent layer of increased specific humidity above a single-layer stratocumulus deck is observed in the period between 5 and 7 June 2017. This measurement case provides the observational basis for this study. For this period, Fig. 2 shows the temporal development of the vertical specific humidity profile derived from radiosonde measurements in combination with cloud boundaries and the time-height curves of the corresponding BELUGA flights. The BELUGA flights were conducted around noon on each of the three consecutive days. A local maximum in specific humidity above the cloud top is observed on

all three days, with a slight diurnal cycle peaking at noon and with a maximum specific humidity around noon on 6 June. It is worth noting that the observations show a well-defined layer of increased specific humidity, rather than a humidity inversion with only a slight decrease above.

     Cloud boundaries in Fig. 2 are estimated from Cloudnet data. The comparison of cloud base height from the ceilometer on-board RV *Polarstern* and corresponding Cloudnet data illustrates that the variability in the cloud boundaries is not represented



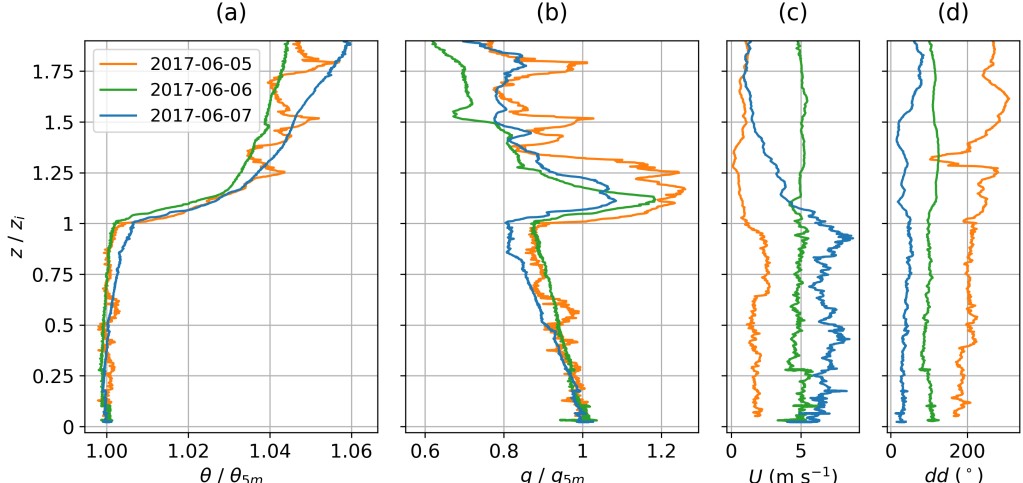

**Figure 3.** Balloon-borne vertical profiles for three noon-time measurement flights on 5, 6, and 7 June 2017. The altitude $z$ is normalized to the temperature inversion base height $z_i$, which is defined as the lower boundary of the temperature inversion layer during the ascent. Potential temperature $\theta$ (a) and the specific humidity $q$ (b) are normalized to their near-surface values. Panels (c) and (d) show horizontal wind velocity $U$ and wind direction $dd$.

in the Cloudnet data, but the temporal development is reproduced sufficiently by both methods. Throughout the three-day period, cloud height and thickness decrease to a minimum at noon of 6 June, and thereafter increase again. The cloud is almost permanently of mixed-phase type with a maximum liquid water content (LWC) between 0.15 $\mathrm{g\,m^{-3}}$ and 0.6 $\mathrm{g\,m^{-3}}$ and an estimated ice water content (IWC) of about 0.03 $\mathrm{g\,m^{-3}}$ derived from Cloudnet data (Griesche et al., 2019a, not shown here).

Figure 3 shows vertical profiles of the three BELUGA measurements during the 5–7 June period. Height is normalized by
the base-height of the temperature inversion. To compare the ABL structure, the potential temperature $\theta$ and $q$ are normalized by their near-surface values (Fig. 3a and 3b). All measurements show a similar vertical structure of $\theta$ and $q$. The ABL below the temperature inversion layer is slightly stably stratified. Above the temperature inversion, the thermodynamic stability is much increased compared to below the inversion. The potential temperature on 5 June exhibits some variations above the inversion. On all days, the cloud is thermodynamically coupled with the surface layer, which manifests in the absence of a temperature
inversion below cloud top.

A distinct humidity layer with slightly varying vertical relative thickness is observed on all days. The relative strength of this layer temporally decreases. The temperature inversion base coincidences with the base of the humidity layer. The vertical profiles of horizontal wind velocity and direction are shown in Fig. 3c and 3d. Throughout the period, the wind velocity increases inside the mixed layer from 2 to 7 $\mathrm{m\,s^{-1}}$ and the wind direction changes from southwesterly to northeasterly. The
wind velocity is almost height-constant on 6 June, whereas on 5 and 7 June it gradually decreases above the mixed layer.

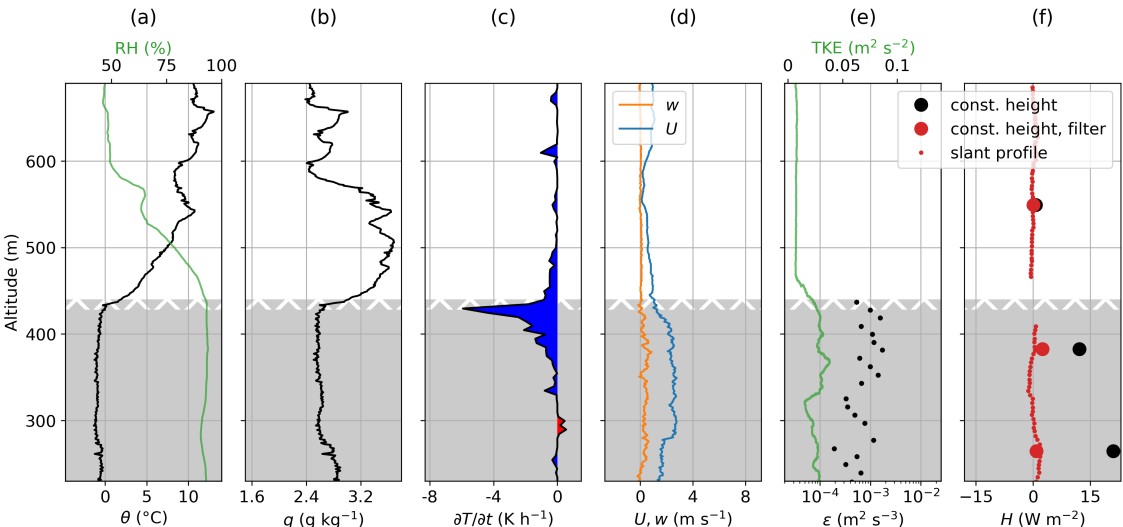

**Figure 4.** Boundary layer observations around cloud top on 5 June 2017: Vertical profiles of (a) potential temperature $\theta$ and RH, (b) specific humidity $q$, (c) terrestrial heating rate $\partial T/\partial t$, (d) horizontal wind velocity $U$ and vertical wind velocity $w$, (e) local dissipation rate $\varepsilon$ and turbulent kinetic energy TKE and (f) virtual sensible heat flux $H$. Small red dots represent flux estimates on the slant profile, big black dots on constant altitude segments. Red big dots represent constant altitude fluxes based on high-pass filtered data. The cloud is shown as shaded area, the cloud top height range as hatched area.

## 3.2 Vertical profiles of mean ABL parameters

For a more detailed study of the humidity layer, measurements close to the cloud top and the temperature inversion are analyzed. Figure 4, Fig. 5 and Fig. 6 a–d each show vertical profiles of mean ABL parameters measured by BELUGA on the 5, 6 and 7 June, respectively. The cloud top height is indicated as a vertical height range instead of a sharp boundary height. Cloud top

variability within this region can be caused by spatial or temporal cloud heterogeneity, and by the use of different measurement methods. The vertical height range shown in the figures expresses the spread in cloud top height among Cloudnet data, RH measurements and the height of maximum radiative cooling, which is supposed to be located at cloud top (Wood, 2012).

Characteristics of the temperature inversion layer and the humidity layer are summarized in Table 1. The strongest temperature difference of 11 K is found on 5 June 2017, decreasing to 7 K for the following two profiles. The temperature inversion

layer depth decreases from 90 m to 40 m and afterwards increases again to 100 m throughout the period. This results in a temperature gradient between 0.07 K m$^{-1}$ and 0.17 K m$^{-1}$, with a maximum on 6 June. The strength of the humidity layer slightly decreases from 1.1 to 0.7 g kg$^{-1}$. Starting with a depth of 150 m on 5 June, a minimum in humidity layer depth of 50 m is observed on 6 June.

On 5 June, the cloud layer is capped by the temperature inversion, whereas the cloud top tends to penetrate slightly into the

inversion on 6 and 7 June. This extension of the cloud layer into the temperature inversion layer is common for Arctic clouds



**Table 1.** Potential temperature inversion (TI) and humidity layer (HL) characteristics for 5, 6 and 7 June 2017.

|  | 5 June | 6 June | 7 June |
|---|---|---|---|
| TI: $\Delta T$ (K) | 11 | 6.8 | 7 |
| TI depth (m) | 90 | 40 | 100 |
| TI gradient (K m$^{-1}$) | 0.12 | 0.17 | 0.07 |
| TI base height (m) | 430 | 290 | 575 |
| HL: $\Delta q$ (g kg$^{-1}$) | 1.1 | 1 | 0.7 |
| HL depth (m) | 150 | 50 | 135 |

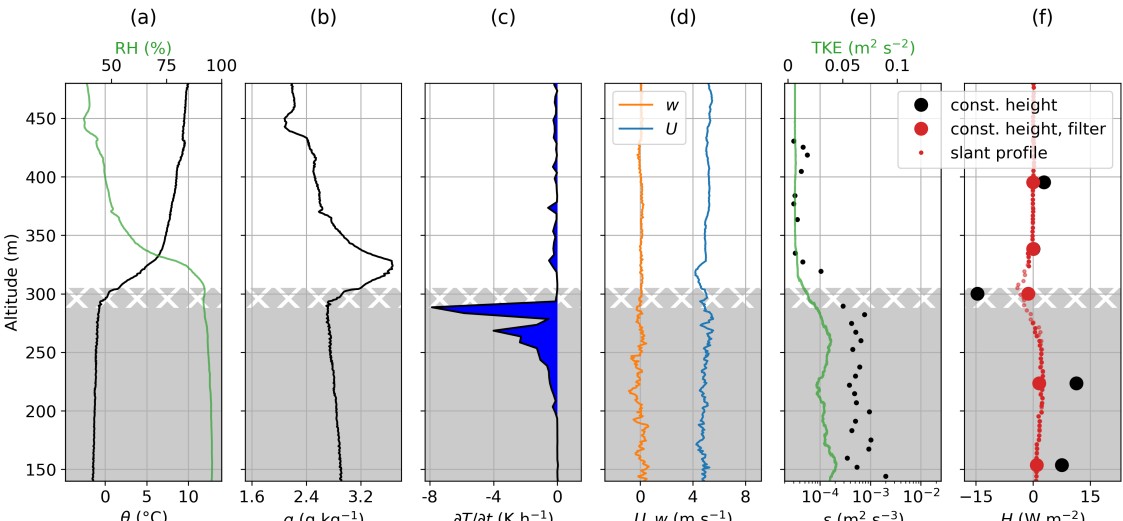

**Figure 5.** Same as Fig. 4, but for 6 June 2017.

(e.g. Pleavin, 2013). Due to the uncertainties in estimating cloud top height, it is challenging to quantify how deep the cloud penetrates into the inversion.

The maximum terrestrial (longwave) radiative cooling (a minimum in the terrestrial heating rate $\partial T/\partial t$) coincidences with the coldest point of the temperature inversion – the inversion base. We observe terrestrial cloud top cooling rates between -6 K h$^{-1}$ and -8 K h$^{-1}$ on all days. In all three cases, the maximum of specific humidity is located clearly above the cloud layer and above the cloud top cooling region.

The transition from the cloud to free troposphere is characterized by a more or less pronounced decrease of horizontal wind velocity, which corresponds to vertical wind shear. An increase in the horizontal wind velocity in the cloud layer is observed on 5 and 7 June. The wind velocity at cloud top decreases by 2 m s$^{-1}$ on 5 June and by 5 m s$^{-1}$ on 7 June. On 6 June there is only a local (20 m vertically extending) decrease in horizontal wind velocity just above the cloud of below 1 m s$^{-1}$. In all cases,


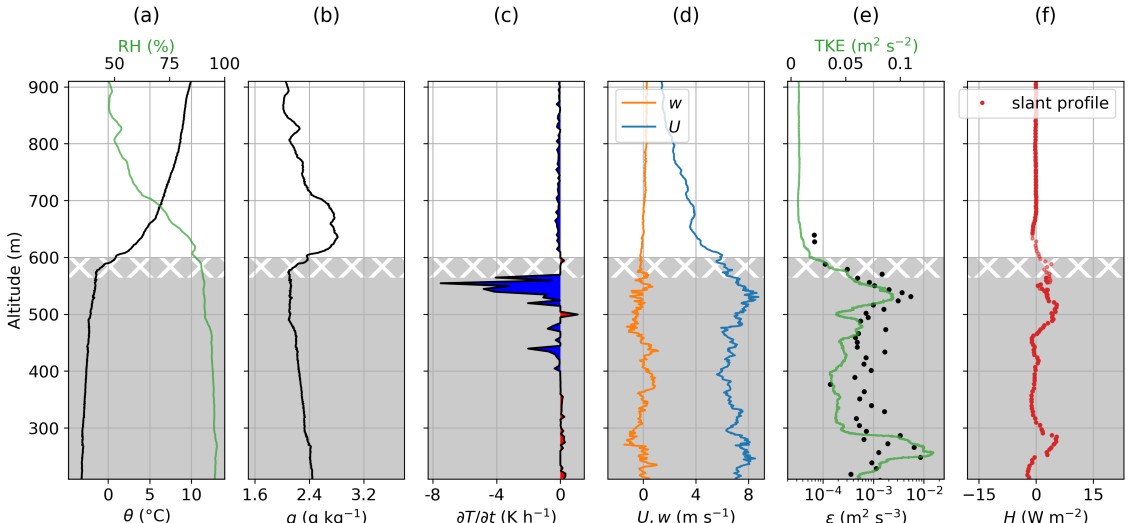

**Figure 6.** Same as Fig. 4, but for 7 June 2017. No constant altitude segments were recorded.

the fluctuations of vertical wind $w$ are pronounced inside the cloud compared to above the cloud. The gradient Richardson number Ri is greater than one for all days (e.g. Ri $\approx 5$ on 6 June), which means that wind shear at cloud top allows only little turbulence.

### 3.3 Turbulent transport from the humidity layer into the cloud

#### 3.3.1 Turbulent ABL parameters

Of key importance for understanding the impact of humidity layers on the underlying cloud layer is to gain insight into the coupling between both layers, as expressed by the vertical transport of humidity between them. To this purpose the profiles of various turbulence-related variables are analyzed. Use is made of a technique to estimate the local dissipation rate $\varepsilon$ and local turbulent kinetic energy TKE from the slant profile data obtained from BELUGA observations, as described in detail by

Egerer et al. (2019b). In the present cases, the local turbulence, as described by $\varepsilon$ and TKE, is most pronounced in the cloud layer for all days with typical values of $\varepsilon \sim 10^{-3}$ m$^2$ s$^{-3}$ and TKE $\sim 0.03$ m$^2$ s$^{-2}$ (Fig. 4e, 5e and 6e). For 7 June (Fig. 6e), with increased wind velocity, a clear maximum of $\varepsilon$ is evident just below cloud top, which is where the terrestrial radiative cooling rate is also at its maximum. At cloud top and within the temperature inversion layer, $\varepsilon$ decreases to the low-turbulent free troposphere level. This transition appears gradual, which shows that the humidity layer is not completely decoupled from

the cloud layer. Instead, both layers are connected by turbulent mixing.





### 3.3.2 Eddy covariance fluxes

To investigate the vertical moisture transport, the vertical turbulent moisture flux (or latent heat flux), which is proportional to $\overline{w'q'}$ needs to be quantified. However, this requires fast humidity measurements, which are not available with our instrumentation. Alternatively, we argue that an estimate of the turbulent virtual heat flux

$$H = \overline{\rho} \cdot c_p \cdot \overline{w'\theta'_v} \tag{1}$$

provides an indication for the direction of the turbulent energy fluxes. Here, $\theta_v$ is the virtual potential temperature (as measured by the ultrasonic anemometer), $\overline{\rho}$ is the mean air density and $c_p = 1005 \, \mathrm{J \, kg^{-1} \, K^{-1}}$ the specific heat capacity of air. Fluctuating parameters are marked with a prime, time averages with an overline. Both temperature and humidity exhibit similarly strong gradients above the cloud layer. Lilly (1968) applied the ratio of the heat flux to the potential temperature difference across the
inversion for defining the entrainment velocity $w_e$:

$$w_e = -\frac{\overline{w'\theta'}|_{z_i}}{\Delta\overline{\theta}} . \tag{2}$$

This relationship can be applied to any conserved variable at the ABL top, such as $q$ (de Roode and Duynkerke, 1997). With a positive $w_e$ and similar $\theta_v$ and $q$ gradients, we can assume that the fluxes of heat and moisture will point in the same direction. This assumption is further discussed in Sec. 3.3.3 and in combination with the LES results.

Vertical profiles of turbulent fluxes can be estimated by (i) measurements averaged over time periods at different constant altitudes, or (ii) by averaging the collected data over altitude segments on a continuous vertical profile (slant profiles) (Egerer et al., 2019b). Slant profiles describe the instantaneous vertical structure of turbulent fluxes, but with a reduced statistical significance of the absolute value. The values strongly depend on the vertical extent of the selected sub-record, the horizontal wind velocity and on the type of implicit high-pass filtering. Another challenge is the high-pass filtering itself in regions of
strong or rapidly changing gradients, which is typical for the temperature inversion height range. In those regions, the choice of the filter type is crucial and can produce artificial fluctuations, which dominate the flux estimate. Therefore, the fluxes in the cloud top region are not shown for the measurements conducted on 5 June. For flux estimates derived from constant altitude flight patterns, the time records are detrended to define the turbulent fluctuations. For the slant profiles, a high-pass filter of Bessel type is applied. The filter window is adjusted to the daily conditions of cloud thickness and horizontal wind, resulting
in a window between $47 \, \mathrm{s}$ and $110 \, \mathrm{s}$. This corresponds to a spatial scale of about $100 \, \mathrm{m}$ to $400 \, \mathrm{m}$, depending on the horizontal wind velocity, and is of the same order as the vertical extent of the cloud layers. Additionally, the same high-pass filter is applied to the constant altitude records for a comparison to the slant profile fluxes.

It is still challenging to obtain vertical profiles of turbulent fluxes. Slant profiles suffer from a short averaging time for flux estimates and omit lower-frequency contributions. Estimating turbulent fluxes on constant altitude segments is statistically
more robust, but more difficult to obtain. Further, measuring in a constant altitude does not ensure stationary conditions because of the high temporal variability of the ABL and the influence of possible gravity waves. It is beyond the scope of this work to discuss all uncertainties of estimating turbulent fluxes (a discussion can be found e.g. in Lenschow et al., 1994). Instead, the presented results should be considered as a qualitative indication of the direction of vertical moisture transport.





During the three-day period of measurements analyzed here, the virtual sensible heat fluxes inside the cloud layers are
predominantly positive (Fig. 4f, 5f and 6f). Values from constant altitude records are up to 20 $\mathrm{W\,m^{-2}}$ for 5 June, and up to
10 $\mathrm{W\,m^{-2}}$ for 6 June. On 7 June, no data from constant altitude measurements are available, but the slant profile fluxes reach
up to 5 $\mathrm{W\,m^{-2}}$ in the cloud. Above the temperature inversion layer, the turbulent fluxes vanish and are close to zero during the
entire three-day period.

On 6 June, a negative virtual sensible heat flux of -15 $\mathrm{W\,m^{-2}}$ is observed close to the cloud top, suggesting vertical mixing
of air from the humidity layer into the cloud top layer. This is the only record with observations at constant altitude inside the
inversion layer, which is hard to locate exactly during the measurements.

The time series for the flux calculation parameters in this altitude are shown in Fig. 7. The temperature record shows rapid
variations of up to 3 K and exhibits structures with a typical time scale of a 30–50 s. The temperature variations are detected
by different sensors (PT100, thermocouple and the Sonic anemometer), which excludes an instrument artefact. The magnitude
of the temperature structures is about half of the temperature difference across the inversion of $\Delta\theta = 6.8$ K.

The temperature fluctuations might result either from the instrument moving up and down in the temperature inversion or
from the air mass oscillating around the instrument. The barometric pressure $p_\mathrm{b}$ spans a range of 0.6 hPa throughout the entire
time series, which corresponds to altitude variations of about 5 m. The measurement altitude for the second part of the record
varies only in the range of $\pm$ 1.5 m. Based on the temperature gradient of 0.17 $\mathrm{K\,m^{-1}}$, this altitude variation corresponds to
a temperature change of $\pm$ 0.3 K. Therefore, the observed temperature structures with amplitudes 20 times higher are mainly
not caused by altitude variations, but instead correlate with changes in the yaw angle of about $10°$, meaning with a change in
wind direction. There are no similar structures in the record of the horizontal wind speed (not shown in Fig. 7).

A more plausible explanation for the observed temperature structures are external gravity waves (as waves may alter the
wind direction), which are frequently observed in temperature inversion layers. Deardorff et al. (1969) observed convectively
induced waves at the interface of a mixed layer and a stable layer in water tank experiments. Further, Driedonks and Tennekes
(1984) reported intermittent turbulence for the entrainment zone of a convective ABL. Gravity waves oscillate with the Brunt–
Väisälä frequency

$$N = \sqrt{\frac{\mathrm{g}}{\theta} \cdot \frac{\partial\theta}{\partial z}}. \tag{3}$$

In our case, this results in $N \approx 0.08$ Hz and a wave period of $2\pi/N \approx 78$ s. This is around three times the observed wave
period of $\sim 25$ to $30\,\mathrm{s}$ in Fig. 7.

The observed temperature structures violate the classical Reynolds decomposition $x'(t) = x(t) - \overline{x}$, which requires $\overline{x'} = 0$
for a sufficient averaging time. However, this is a major precondition for the flux calculation. The calculated amplitudes of
the temperature fluctuations strongly depend on how the averaging is carried out. In Fig. 7, $\theta'_\mathrm{v}$, $w'$ and $\theta'_\mathrm{v}w'$ are calculated by
subtracting the linear trend (black curves) and by high-pass filtering with a Bessel filter with a filter time window of 47 s (red
curves). The filter parameters are the same as applied for filtering on the slant profile. The results for the fluxes estimated from
filtered time series observed at constant heights are added as red dots in Fig. 4f and 5f. When applied to all constant altitude



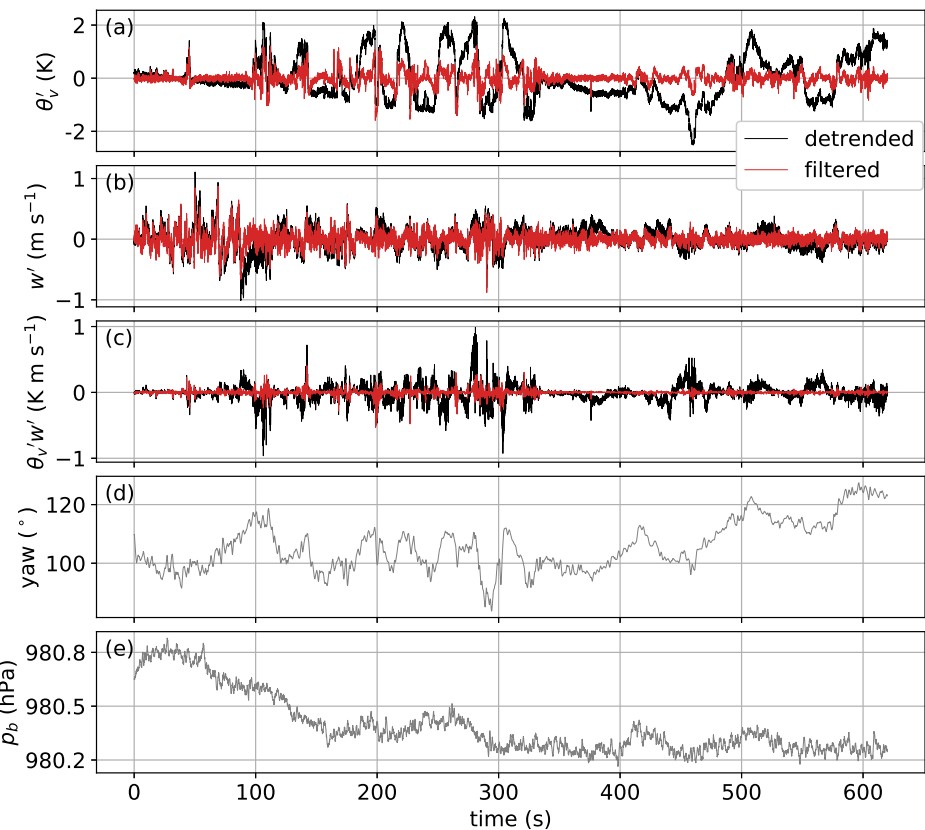

**Figure 7.** Time series of (a) virtual potential temperature $\theta_\mathrm{v}$, (b) vertical wind $w$ and (c) covariance $\theta_\mathrm{v}{}'w'$, (d) yaw angle and (e) barometric pressure $p_\mathrm{b}$. The data are recorded on 6 June 2017 on the constant altitude segment in 302 m±3 m altitude and used for the flux calculation in Fig. 5.

time series, the filter algorithm reduces the magnitude of the estimated fluxes, but the vertical structure remains. Magnitude and vertical structure are comparable to the results of the slant profile.

The cloud top virtual sensible heat flux on 6 June results mainly from temperature fluctuations, rather than from wind velocity fluctuations. Independent from the filter settings, the virtual sensible heat flux estimate for the cloud top region is negative. This agrees with the sign of the flux estimate on the slant profile of $H \approx -3.5 \ \mathrm{W\,m^{-2}}$ at cloud top. Because the temperature and specific humidity gradients have the same sign, the moisture flux and the measured virtual sensible heat flux should be orientated in the same direction. We conclude, that the negative cloud top flux on 6 June suggests entrainment of humid air from above into the cloud.





On 5 June, the missing flux estimates close to cloud top do not allow for a conclusion about entrainment. On 7 June, only a weak negative virtual heat flux of -1 $\mathrm{W\,m^{-2}}$ is measured around 50 m above the cloud top on the slant profile. Therefore, no statement can be made about entrainment for the measurements conducted on 5 and 7 June 2017.

### 3.3.3   Gradient method fluxes

One key issue concerning the observed humidity layers above cloud top is their interaction with the cloud layer via vertical
moisture transport. Due to limitations of the BELUGA instruments (e.g. the lack of a fast response humidity sensor), only the virtual sensible heat flux $H$ can be directly estimated by applying the eddy covariance method. Based on turbulence parameters in combination with the mean profiles of $q$ and $\theta_\mathrm{v}$, a rough estimate (at least of the sign) of the moisture flux is possible in order to draw conclusions about the vertical moisture transport. The $q$ and $\theta_\mathrm{v}$ fluxes are additionally influenced by the liquid water flux (de Roode and Duynkerke, 1997; Nicholls, 1984). Here, we neglect the effect of cloud water phase transition at cloud top
and assume the $q$ and $\theta_\mathrm{v}$ fluxes to be independent from each other.

    Based on the classical boundary layer theory (e.g. Stull, 1988), the turbulent energy fluxes are related to the vertical mean gradients of the respective parameter $x$ by:

$$\overline{w'x'} = -K_\mathrm{x} \cdot \frac{\partial \overline{x}}{\partial z} \tag{4}$$

with $K_x \geq 0 \ \mathrm{m^2\,s^{-1}}$ being the turbulent exchange coefficient. This relationship does not consider counter-gradient fluxes,
which, however, should be negligible in strong inversions. For heat and moisture, it has been shown that $K_\mathrm{H} \approx K_\mathrm{Q}$ (Dyer, 1967) for a wide range of stratification. We cannot provide a direct estimate for $K_\mathrm{H}$ or $K_\mathrm{Q}$ to apply the gradient method. Instead, we can estimate the exchange coefficient for momentum $K_\mathrm{m}$: Hanna (1968) suggested

$$K_\mathrm{m} = \mathrm{C} \cdot \frac{\sigma_\mathrm{w}^4}{\varepsilon} \tag{5}$$

with C = 0.35. The required basic turbulence observations $\sigma_\mathrm{w}$ and $\varepsilon$ are available from measurements obtained with the BEL-
UGA instrument setup. Finally, the turbulent Prandtl number $\mathrm{Pr_t} = K_\mathrm{m}/K_\mathrm{H}$ relates the exchange coefficients of momentum and heat. The Prandtl number is a function of vertical thermodynamic stability, but ranges from 0.5 to 1 (Li, 2019). Here, we consider $\mathrm{Pr_t} \approx 0.7$ as suggested by Stull (1988), although there is a controversial discussion about $\mathrm{Pr_t}$ in particular for stable conditions such as the inversion layer (e.g. Grachev et al., 2007).

    For the observations performed on 6 June 2017, we estimate from the horizontal BELUGA flight leg at cloud top $\sigma_\mathrm{w}^2 \approx$
$0.03\,\mathrm{m^2\,s^{-2}}$ and $\varepsilon \approx 4 \cdot 10^{-4}\,\mathrm{m^2\,s^{-3}}$, yielding $K_\mathrm{m} \approx 0.95\,\mathrm{m^2\,s^{-1}}$ and $K_\mathrm{H} = K_\mathrm{Q} \approx 1.35\,\mathrm{m^2\,s^{-1}}$. The values estimated from the slant profile are by a factor of almost five smaller yielding $K_\mathrm{H} = K_\mathrm{Q} \approx 0.28\,\mathrm{m^2\,s^{-1}}$. In the following, we use the values derived from the slant profile for convenience; a quantitative description of the wave influence is not possible at this point.

    With a vertical temperature gradient of $0.18\,\mathrm{K\,m^{-1}}$ (estimated for the hatched height range in Fig. 5a), we calculate:

$$H = -\overline{\rho} \cdot \mathrm{c_p} \cdot K_\mathrm{H} \cdot \frac{\partial \overline{\theta}_\mathrm{v}}{\partial z} \tag{6}$$

and obtain $H \approx -59\,\mathrm{W\,m^{-2}}$, which is four times higher compared to the direct covariance estimate ($H = -15\,\mathrm{W\,m^{-2}}$). For the same height range in Fig. 5b, the vertical specific humidity gradient is estimated to $0.025\,\mathrm{g\,kg^{-1}\,m^{-1}}$ and the latent heat





**Table 2.** Turbulence parameters (vertical wind variance $\sigma_w^2$, dissipation rate $\varepsilon$ and turbulent exchange coefficients $K_m$, $K_H$ and $K_Q$) and turbulent fluxes ($H$ and $L$) estimated from the gradient method on the slant profiles on 6 and 7 June 2017

|  | 6 June | 7 June |
|---|---|---|
| $\sigma_w^2\,(\mathrm{m^2\,s^{-2}})$ | $10^{-2}$ | $5\cdot10^{-3}$ |
| $\varepsilon\,(\mathrm{m^2\,s^{-3}})$ | $2\cdot10^{-4}$ | $10^{-4}$ |
| $K_m\,(\mathrm{m^2\,s^{-1}})$ | 0.19 | 0.09 |
| $K_H = K_Q\,(\mathrm{m^2\,s^{-1}})$ | 0.28 | 0.13 |
| $H\,(\mathrm{W\,m^{-2}})$ | -59 | -24 |
| $L\,(\mathrm{W\,m^{-2}})$ | -21 | -6 |

flux

$$L = -\overline{\rho} \cdot L_\mathrm{v} \cdot K_Q \cdot \frac{\partial \overline{q}}{\partial z} \tag{7}$$

with the latent heat of evaporation $L_\mathrm{v} = 2.5 \cdot 10^6\,\mathrm{J\,K^{-1}}$ yields $L \approx -21\,\mathrm{W\,m^{-2}}$.

For the case observed on 7 June, the turbulent exchange between cloud top and inversion layer is less pronounced and with $K_H = K_Q \approx 0.13\,\mathrm{m^2\,s^{-1}}$ we estimate $H \approx -24\,\mathrm{W\,m^{-2}}$ and $L \approx -6\,\mathrm{W\,m^{-2}}$, respectively. The fluxes are weaker because of the higher variance and dissipation rate prevailing on 6 June, although the 7 June case exhibits stronger wind shear and less stability close to the cloud top. All turbulence parameters observed during the slant profile BELUGA flight tracks are summarized in Table 2. The gradient method is not applied to the 5 June results, because there is hardly any measurement of

dissipation in the inversion layer due to the resolution of the Sonic anemometer. For the measurement cases on 6 and 7 June, the observations consistently show negative latent and virtual sensible heat fluxes at cloud top, meaning a downward flux of sensible and latent heat from the humidity layer into the cloud top. This negative sign is a direct consequence of the positive gradients of potential temperature and moisture above cloud top.

## 4 Large Eddy Simulations (LES)

To further investigate the turbulent transport processes between the SHI and the cloud top, the in situ measurements are complemented with LES. The LES provides the turbulent flux of humidity in the vicinity of the inversion, which gives information about the interaction of the humidity layer with the underlying mixed layer.

### 4.1 Model configuration

The LES configuration adopted in this study was designed by Neggers et al. (2019) for the PASCAL observation period 5–7
June 2017. The Dutch Atmospheric Large-Eddy Simulation model (DALES, Heus et al., 2010) is applied and equipped with a well-established double moment mixed-phase microphysics scheme (Seifert and Beheng, 2006). A Lagrangian framework is



adopted following evolving cloudy mixed layers in warm air masses as they moved towards the RV *Polarstern*. The simulated doubly periodic domains are discretized at 10 m vertical and 20 m horizontal resolution, while the large-scale forcing is derived from analysis and forecast data of the European Centre for Medium-range Weather Forecasts (ECMWF). Surface temperature is

prescribed, while the surface fluxes are interactive, resulting in weakly coupled cloudy mixed layers. The temperature inversion height $z_i$ and cloud layer boundaries are free to evolve. The simulations are constrained by in situ radiosonde profiles and evaluated against further independent cloud measurements. Eight cases are constructed during the three-day study period, capturing the variation in cloud and thermodynamic properties observed during this period.

The PASCAL simulations described above are thoroughly evaluated against measurements. Although in general the LES

reproduces these to a satisfactory degree and also does produce humidity inversions, their strength and depth is underestimated. For this reason additional simulations are performed for this study, designed to better represent the observed humidity layers on 6 and 7 June 2017. The configuration of these new simulations differs from the setup described above in three aspects:

- Instead of starting two days in advance, the model initializes only 12 hours before the arrival of the Lagrangian air parcel at RV *Polarstern*. A shorter lead time allows to adjust the initial conditions such that a good agreement is obtained with

the BELUGA sounding in terms of temperature inversion height. On the other hand, a period of 12 hours is still long enough to allow complete spin-up of the mixed-phase clouds.

- The initial state adjustments include a lowering of the inversion height, following the method of Neggers et al. (2019). However, in addition a humidity layer of 200 m depth and 0.5 g kg$^{-1}$ strength is superimposed on the initial profile, placed immediately above the new temperature inversion. These values reflect the structure of the observed SHIs.

- The surface sensible and latent heat fluxes are switched off, in effect decoupling the cloud layer from the surface. Imposing a surface decoupling has proven to be an effective way to maintain humidity inversions (Solomon et al., 2014). It should be noted that no measurements were made of the surface heat fluxes along the upstream trajectory, preventing us from assessing the validity of this modification.

These modifications make the case slightly idealized, but are justified by our goal of working with an LES realization in

which the strength and depth of the humidity layers more or less matches the BELUGA observations. This is prerequisite for using LES data alongside BELUGA data for studying humidity inversion processes such as turbulent fluxes. For reference, the simulations are repeated for an initial state without the SHI superimposed. Because the method described above works best for the 7 June case, the results for this day are discussed below. The 6 June case is briefly touched, but the results exhibit some numerical artefacts, which are outlined in Appendix A.

**4.2  Comparison of LES and BELUGA results**

Figure 8 shows vertical profiles of the LES output and of BELUGA measurements (Fig. 6) for 7 June 2017. The LES profiles are results of the 12 h simulations, ending at the location of RV *Polarstern* at 10:48 UTC. Here, all variables represent horizontal averages over the full LES domain (2.56 km × 2.56 km) and are averaged over 900 s. This includes the turbulent fluxes of

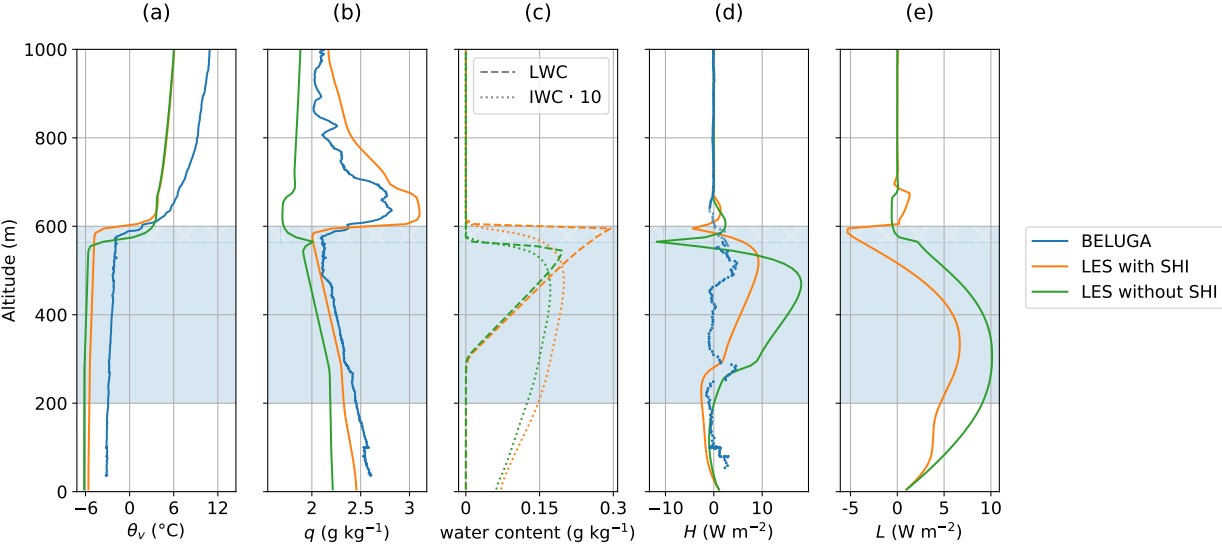

**Figure 8.** Comparison of LES results (with and without an initial SHI) and BELUGA observations for 7 June 2017: Vertical profiles of (a) virtual potential temperature $\theta_v$, (b) specific humidity $q$, (c) liquid (LWC) and ice water content (IWC) of the LES, (d) virtual sensible heat flux $H$ and (e) latent heat flux $L$. The light blue area is the cloud extent for the observations (cloud top is derived as in Sect. 3 for BELUGA, cloud base is from Cloudnet).

heat $H$ and moisture $L$, calculated as the covariance between vertical velocity and perturbations in static energy and humidity, respectively. The LES results are shown for simulations with and without an initial SHI.

With an initial SHI in the LES, the temperature inversion base, and therefore the mixed layer height, agrees well with the observed inversion base (Fig. 8a). Without the initial humidity layer, the temperature inversion base is around 40 m lower. A constant temperature offset of around 2 K between measured and simulated profiles is present, which is due to the initial LES profile based on a radiosounding. The temperature difference across the inversion as well as temperature gradients are comparable. The vertical profile of specific humidity shows a similar vertical structure and a distinct increase of $q$ above the cloud layer in both the model and the observations (Fig. 8b). The strength of the SHI of $\Delta q = 1.1\ \mathrm{g\,kg^{-1}}$ in the LES is close to the observed humidity inversion strength of $\Delta q = 0.7\ \mathrm{g\,kg^{-1}}$. In the LES without initial SHI, specific humidity decreases by $\Delta q \approx 0.2\ \mathrm{g\,kg^{-1}}$ within the temperature inversion height range.

Compared to the balloon measurements, a thinner liquid cloud layer forms in the LES, as indicated in the LWC profiles in Fig. 8c. While the observed cloud was around 400 m thick, the simulations result in a liquid cloud of about 300 m vertical extent. Note that significant ice water is present below the liquid cloud base in the model, for which ceilometer readings are sensitive (Bühl et al., 2013). For this reason the model bias in cloud base height could be artificial. Without a humidity layer, the liquid cloud is slightly thinner, extending only 260 m. The cloud top is simulated at around 600 m altitude for the scenario with SHI and at 560 m altitude for the scenario without SHI, respectively. In both cases, the cloud top is slightly (10–20 m)



above the temperature inversion base. In the SHI case, the higher cloud top reflects the larger mixed layer depth compared to the case without SHI.

The BELUGA measurements performed on 7 June provide sensible heat flux measurements on the slant profile, which means that flux magnitude is underestimated and allowing only a qualitative comparison to the LES. Both BELUGA and LES provide a positive virtual sensible heat flux inside the cloud layer, increasing with altitude (Fig. 8d). In the LES, a noteworthy

feature is the negative virtual heat flux at cloud top, which is seen with and without initial SHI (-5 $\mathrm{W\,m^{-2}}$ and -12 $\mathrm{W\,m^{-2}}$, respectively). In the BELUGA observations, a small negative virtual sensible heat flux of -1 $\mathrm{W\,m^{-2}}$ is present around 50 m above cloud top. Unfortunately, the BELUGA instrumentation does not provide moisture flux measurements. The LES, with or without an initial SHI, shows a positive (i.e. upward directed) moisture flux $L$ between surface and cloud top with a maximum at cloud base (Fig. 8e). In the presence of an initial SHI, the cloud top region exhibits a negative moisture flux of -5 $\mathrm{W\,m^{-2}}$.

This negative moisture flux coincidences with the negative virtual sensible heat flux, and indicates that downward humidity transport takes place between the humidity layer and the underlying mixed layer. Lacking the initial SHI, the total moisture flux is close to zero near the inversion. This means that in this case dry air, rather than humidity, is entrained into the mixed layer from above. In Sect. 3.3 we argue that the cloud top virtual sensible heat flux might be an indicator for the direction of the moisture flux. This is in agreement with the LES results for 7 June with an SHI, where both fluxes point into the same

direction, whereas the LES without SHI shows a negligible moisture flux.

For the 6 June simulations (Fig. A1), the LES with SHI matches the observations well in terms of temperature inversion height, but the vertical profile of $q$ is not as well represented as for 7 June. The impact of removing the initial SHI is the same as for 7 June: The temperature inversion stays around 40 m lower, and the liquid cloud layer remains thinner (230 m thick compared to 300 m thick with initial SHI). At cloud top, a negative $H$ is visible with and without SHI (-11 and -10 $\mathrm{W\,m^{-2}}$,

respectively). With SHI, at cloud top a weak negative $L$ of -0.3 $\mathrm{W\,m^{-2}}$ is present, whereas without SHI there is no significant negative peak at cloud top. Although for $L$ the simulations show some numerical artifacts inside the SHI (as explained in Appendix A), the main conclusion for the 6 June case is the same as for 7 June: The humidity layer provides moisture for the cloud, manifested by a downward moisture flux at cloud top.

## 5   Discussion

### 5.1   Formation of specific humidity inversions

Before exploring the interactions of the SHIs with clouds, we briefly discuss their origin. Mostly, the formation of Arctic summertime SHIs is attributed to large-scale advection of humid air masses (Solomon et al., 2014; Naakka et al., 2018). Tjernström et al. (2019) developed a conceptual model for air mass transformation during moist and warm air advection over open water and sea ice: When warm, humid air is advected over sea ice, the air mass is cooled by the surface, a surface

temperature inversion develops and fog or low clouds form. Cloud top buoyancy and surface roughness enhance mixing, the mixed layer deepens and the clouds lift from the surface. The last stage is the well-mixed, cloud-capped, persistent ABL, which has been observed frequently in the Arctic. During the air mass transformation, the cooling within the ABL results

**Figure 9.** Five-day back trajectories ending at RV *Polarstern* (gray dot) at 00:00 and 12:00 UTC between 5 and 7 June 2017 in altitudes 50m (red), 250m (blue) and 1000m (green). The trajectories are calculated using HYSPLIT (Stein et al., 2015). The LES trajectories for 6 and 7 June (on the 950 hPa isobar) are added as black lines. Sea ice data are for 6 June 2017 (Maslanik and Stroeve, 1999).

in condensation and subsequent precipitation, which reduces the specific humidity. Specific humidity above the ABL is not affected and, as a result, an SHI at the top of the ABL forms.





Figure 9 shows back trajectories of the air masses for the study period between 50 m and 1000 m altitude. The trajectories are calculated using the Hybrid Single-Particle Lagrangian Integrated Trajectory model (HYSPLIT;  Stein et al., 2015). The air mass during the study period originates further south in the Arctic ocean and is advected over open water and sea ice. Towards the end of the period, the air mass resides locally over the sea ice after being advected, which is also seen in the change in wind direction (Fig. 3d). The HYSPLIT trajectories compare well with the 12-hour trajectories used for the Lagrangian LES study, 

which are based on analysis and forecast data of the Integrated Forecast System (IFS) of the ECMWF model.

   If the conceptual model of Tjernström et al. (2019) is applied to the air mass history in the study period, the state at RV *Polarstern* corresponds to the final state with an elevated temperature inversion above the mixed layer and a cloud deck. The formation of the humidity layer is probably a result of warm air advection over sea ice. It remains open why the observations do not only show an SHI, but instead a well-defined layer of increased humidity above the cloud. One reason might be vertically 

differential advection.

## 5.2   Cloud top fluxes

The virtual sensible heat flux at cloud top is estimated with the direct eddy covariance method on constant altitude legs and on the slant profile. This is complemented by applying the gradient method to the observations in order to add the latent heat flux. For both methods, the fluxes on the slant profile are somewhat smaller than on the constant altitude legs (for the 6 June case 

a factor of four to five). There are two possible explanations for this difference: (i) the values derived from the horizontal legs are more reliable and statistically more robust because more eddies of the typical integral length scale $\mathcal{L} \propto \sigma^3/\varepsilon \approx$ 6–15 m (Wyngaard, 2010) have been sampled, and (ii) gravity waves (see discussion in Sect. 3.3.2) may also influence the estimates of the variance and dissipation and, therefore, the turbulent exchange coefficients.

   The fluxes based on the observations on 6 and 7 June can be compared to the LES fluxes. The cloud top flux magnitudes 

from the gradient method (Table 2) differ from the LES fluxes (Fig. 8 and Fig. A1) by a factor of up to five for the most part. For 6 June, the LES cloud top virtual heat flux of -11 W m$^{-2}$ is close to the direct observational estimate of -15 W m$^{-2}$. In summary, all cloud top fluxes (from both observational flux methods and the LES) on 6 and 7 June are consistently negative. Further, the LES provides $K_{\mathrm{H}}$ from the LES fluxes and the vertical temperature gradient by applying Eq. (6). The values for the LES ($K_{\mathrm{H}} \approx 0.03$ m$^2$ s$^{-1}$ for 6 June and $K_{\mathrm{H}} \approx 0.02$ m$^2$ s$^{-1}$ for 7 June) are one order of magnitude lower than the observed 

values of 0.28 m$^2$ s$^{-1}$ and 0.13 m$^2$ s$^{-1}$.

   However, a day-to-day comparison and a comparison between LES and observations should not be over-interpreted. Applying both observational flux methods on one single profile results in an instantaneous flux estimate compared to area- and time-averaged fluxes as done for the LES. Further, the observed fluxes on constant altitude legs are averaged over a limited time period. From a statistical point of view, the three methods provide comparable results only if the integral time scales $\tau$ are 

small compared to the sampling time $T_{\mathrm{s}}$. The relative error of the flux measurement due to the random nature of the temperature and vertical velocity field is given by:

$$\Delta H = \sqrt{\frac{2\tau}{T_{\mathrm{s}}}} \tag{8}$$





(Lumley and Panofsky, 1964). For the slant profile method on 6 June, we integrate the measurements over $T_s = 50$ s and estimate an integral time scale $\tau = \frac{\mathcal{L}}{U} = \frac{6\,\mathrm{m}}{5\,\mathrm{m\,s^{-1}}} = 1.1$ s, yielding a relative error of $\Delta H \approx 20\%$. For the leg at constant height ($\tau = \frac{15\,\mathrm{m}}{5\,\mathrm{m\,s^{-1}}} = 3$ s), a 10 min long sample was taken reducing $\Delta H$ to $\sim 10\%$. Note that this is only the random sampling error ignoring the possible influence of sensor noise, the influence of gravity waves or violations of a homogeneous and stationary sample.

This approach does not allow to directly compare the results for the eddy covariance method, the gradient method and the LES. Instead, it shows that an instantaneous measurement can deviate significantly from averaged samples. Nevertheless, the main ABL feature results from all methods: consistent negative fluxes of virtual sensitive heat and latent heat at cloud top, indicating entraiment of humid air into the cloud.

### 5.3 Influence of the humidity layer on cloud lifetime

The LES study by Neggers et al. (2019) investigates remote and local controls on the mixed layer evolution. Amongst others, it is concluded that gradual and continuous entrainment deepening of the mixed layer (a local process) is observed as long as a cloud is present. Near the temperature inversion, large-scale vertical advection (subsidence and upsidence) is the main control of deepening or shallowing.

Strong sudden subsidence events may cause cloud collapse with a rapid decrease in mixed layer depth. The Cloudnet observations show that the cloud layer on 6 June shallows and descends (Fig. 2), but does not decay although subsidence is present. While the SHI strength is generally underestimated in the original LES, the larger observed SHI strengths may be responsible for maintaining the mixed layer thickness by supplying enough moisture for entrainment processes to prevent cloud collapse.

This is supported by the 12 h trajectories used for the LES presented in Sect. 4, which exhibit a thinner cloud layer when no initial SHI is present. Further, the simulations show that the SHI is responsible for an elevated mixed layer height. Less terrestrial radiative cooling due to a reduced cloud top LWC implies a lower entrainment rate, so that the inversion rises less quickly. The fact that this impact is the same for the 6 and 7 June case, although the cases are different (6 June with subsidence, while 7 June exhibits upsidence), makes this a robust result. More research is necessary to further investigate how the additional entrained moisture of the humidity layer is processed in the cloud (e.g. through phase transition) and how exactly the humidity layer contributes to the cloud evolution (e.g. the role of clouds penetrating into the inversion or thermodynamically decoupled clouds).

### 6 Summary and conclusions

In this study, layers of increased specific humidity (so-called specific humidity inversions, SHI) above Arctic stratocumulus and their interactions with the underlying cloud layer are investigated by means of tethered balloon-borne observations in the north-west of Svalbard ($82°$ N, $10°$ E). A persistent layer of increased specific humidity above the stratocumulus deck is observed and analyzed in detail over a three-day period from 5 to 7 June 2017. The observational data is supplemented with results from dedicated LES experiments that are based on field campaign data.





Local ABL parameters (temperature, humidity, wind, terrestrial irradiance and eddy dissipation) are sampled by in situ measurements with high-resolution instruments for collocated turbulence and radiation observations. Typically, the sampling strategy of the balloon observations is based on continuous vertical profiling combined with short ($\approx 10$ min) flight legs at constant height allowing for statistically significant sampling of turbulence parameters. It turns out to be challenging to position the balloon for a longer time inside a shallow temperature inversion layer above cloud top to sample under homogeneous
and stationary conditions. This is partly caused by the usually non-stationary cloud top height and the varying height of the balloon itself (although this is a minor issue for this study), which causes significant uncertainties in a region with strong gradients. Furthermore, non-turbulent features such as gravity waves in the temperature inversion violate classical Reynolds decomposition, resulting in further uncertainties of the estimated turbulent fluxes.

Although partly limited due to specific measurement conditions, direct turbulent flux measurements in the temperature and
humidity inversion layer are presented and compared with gradient method estimates. For the measurement case observed on 6 June 2017, direct flux estimates at cloud top yield a negative virtual sensible heat flux, which indicates downward entrainment of humid air from above into the cloud layer. The absolute value of the fluxes remains uncertain due to the possible influence of gravity waves and necessary filtering, but – together with gradient flux estimates and LES – a negative latent heat flux of -5 to -20 W m$^{-2}$ is derived. We conclude that the increased humidity above the cloud provides the necessary moisture to sustain
the cloud layer in the observed case.

LES are performed to complement the observations on 6 and 7 June. The simulations confirm the negative virtual sensible heat fluxes and moisture fluxes at cloud top. Simulations without an initial SHI do not show the downward moisture flux at cloud top. Further, analyses of the LES reveal that the SHI is responsible for a slightly thicker cloud layer than in cases without SHI. In addition, the presence of a humidity layer causes an elevated inversion base and cloud layer, which might contribute to
sustain the cloud.

The new BELUGA tethered balloon system proved its unique capabilities to examine the turbulent structure of cloud-topped Arctic boundary layers. These observations motivate further detailed investigations of SHIs and their influence on cloud layers. Improvements of the instrumentation, such as including fast humidity sensors and cloud microphysical observations, will allow for a more detailed insight into the cloud top processes. It is not yet clear how exactly the additional moisture above cloud top
influences the cloud development and lifetime. This question can be answered by future extended observations in combination with more detailed model studies.

## Appendix A:  LES for 6 June 2017

Figure A1 shows LES results for the 6 June 2017 case. While the LES successfully reproduces an SHI, it is less pronounced as in the 7 June case (see Fig. 8). Perhaps this is due to the SHI being much shallower on this day; as a result, it is resolved less
well in the LES, and probably overly weakened by subgrid diffusion. A few features in the $L$ profile capture the eye that are probably related:





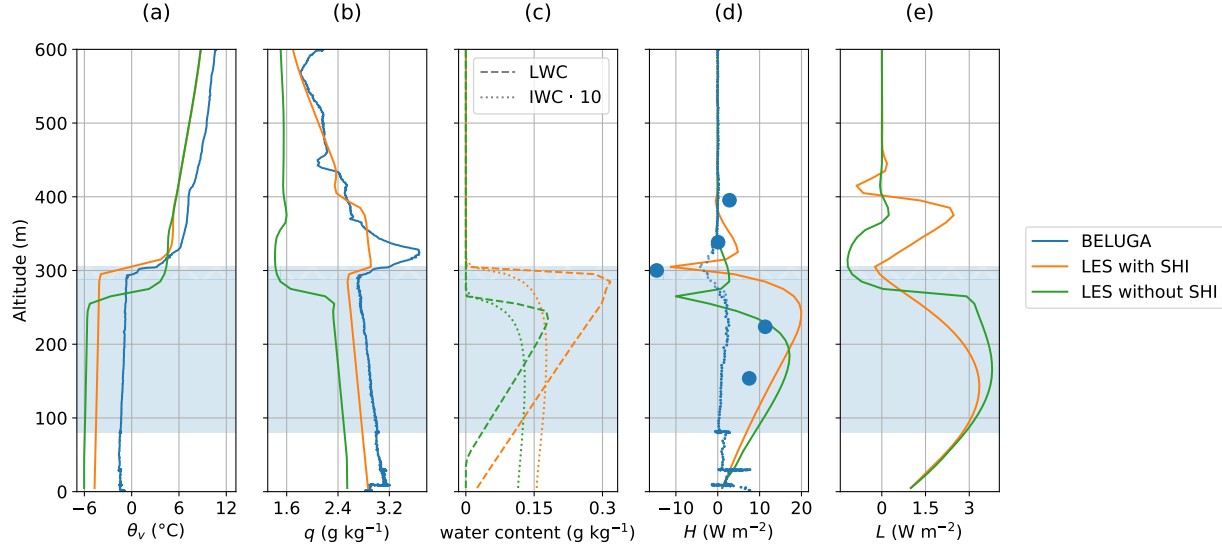

**Figure A1.** Same as Fig. 8, but for 6 June 2017. The blue dots in panel (d) represent the eddy covariance flux estimate on constant altitude legs for BELUGA.

– The simulation including an SHI (orange) shows a positive peak in $L$ in the middle of the humidity layer. This peak suggests that weak turbulent overturning takes place inside that layer. Indeed, in Fig. A1(a) the simulated SHI looks too internally well-mixed, and lacks the shallow but strong peak as observed.

– The simulation without an SHI (green line) has a pronounced negative peak in $L$ above the LES inversion, although not at the inversion. This dip collocates with a dip in the $q$ profile (panel b).

Both features are also present in the 7 June case but much less pronounced. The dip in the flux above the thermal inversion is a well known shortcoming of centered difference advection schemes (e.g. Stevens et al., 2001), and its presence here immediately suggests that this is a numerical artefact. It is probably aggravated by the thinness of the SHI in this case, making it less resolved 485 and thus more sensitive to the choice of the advection scheme. The enhanced internal overturning in the simulated SHI is very likely also related to this issue. However, note that the impact of the SHI on the cloud layer is still consistent with the 7 June case, elevating the temperature inversion and enhancing liquid cloud mass. For this reason we conclude that this numerical artefact does not significantly harm the main conclusions of this study, namely that the humidity layer acts as a moisture source for the clouds below. We will further investigate this numerical problem in the future, by applying higher vertical resolutions 490 and testing different advection schemes in the LES.



*Data availability.* Data related to the present article are available open access through PANGAEA – Data Publisher for Earth & Environmental Science: https://doi.pangaea.de/10.1594/PANGAEA.899803 (Egerer et al., 2019c) and https://doi.pangaea.de/10.1594/PANGAEA.899233 (Egerer and Siebert, 2019a).

*Author contributions.* UE and MG performed the measurements and analyzed the observational data. HS was responsible for the overall
balloon system. HS, MW and AE contributed to the data analysis. RN performed the LES and analyzed the results. UE drafted the paper with contributions from all co-authors.

*Competing interests.* The authors declare that they have no conflict of interest.

*Acknowledgements.* We gratefully acknowledge the funding by the Deutsche Forschungsgemeinschaft (DFG, German Research Foundation) - project number 268020496 - TRR 172, within the Transregional Collaborative Research Center "ArctiC Amplification: Climate Relevant
Atmospheric and SurfaCe Processes, and Feedback Mechanisms (AC)[3]" in sub-project A02. We greatly appreciate the participation in RV *Polarstern* cruise PS 106.1 (expedition grant number AWI-PS106-00). We thank ECMWF for providing access to the large-scale model analyses and forecasts fields used to force the LES. We gratefully acknowledge the Regional Computing Centre of the University of Cologne (RRZK) for granting us access to the CHEOPS cluster. The Gauss Centre for Supercomputing e.V. (www.gauss-centre.eu) is acknowledged for providing computing time on the GCS Supercomputer JUWELS at the Jülich Supercomputing Centre (JSC) under project no. HKU28.
The authors gratefully acknowledge the NOAA Air Resources Laboratory (ARL) for the provision of the HYSPLIT transport and dispersion model used in this publication.



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
