# Peer review of "Case study of a humidity layer above Arctic stratocumulus and potential turbulent coupling with the cloud top"

_Atmospheric Chemistry and Physics, 2020_

## Referee Comment (RC1) · Anonymous Referee #1 · 23 Jul 2020

**Case study of a humidity layer above Arctic stratocumulus using balloon-borne turbulence and radiation measurements and large eddy simulations**
by
Egerer, Ehrlich, Gottschalk, Neggers, Siebert & Wendisch
Manuscript # acp-2020-584
* * *
*Introduction*

With an excessively long and tedious title, this study focuses on so-called moist inversions capping Arctic stratocumulus, starting with observations and extending the discussion with both LES and trajectory calculations. The authors are blessed with probably the only extensive observational dataset on particular situation that exist, and this work has potential to become really groundbreaking. Unfortunately, the manuscript does not capitalize on this possibility; instead it gets bogged down by uncertainty and what could not be done because of various difficulties.

I'm quite sure the authors are sitting on a gold mine here; unfortunately, they don't seem to know what tool to use to excavate it. As read on, my thoughts gradually shift from minor to major revision, but with lots of detailed comments. However, having read to the end I find myself recommending that the manuscript is *rejected*, but with a *strong recommendation* to the authors not to give up, but to explore other avenues of analysis to unlock some of the mysteries behind these cases.

I have many detailed comments that I would love to share with the authors if and when I get a better organized revised manuscript; below are a few overriding concerns.

*General comments*

Analyzing case-studies like this is difficult; even if you find something, it's hard to generalize, and when you find a technique to analyze the data there isn't much more data to test the generality of your method. This is like learning an art; you need to develop a sense of what is working and what is doomed to fail and you need to be ingenious and imaginative. I've been doing this for many years, and I still learn new tricks and its wonderful when finding that data suddenly makes sense!

Here, the authors seem lost; they don't seem to know – or at least they don't tell me – what they are really looking for, they have no map telling them how to get where they want to be, and they don't know what to do next. There is no clear hypothesis to test and even if there was, they don't seem to know how. The paper reads like a chronological description of how the work was progressing, and it probably is. I want a coherent narrative; a story focusing on the science.

After reading a while, it starts to appear like one goal is to find at least a portion of one flight from which they can get a measure of the downward flux of moisture. I fail to understand why knowing the size of that flux – from one case – is so important. But they have a lot of different data to analyze in different ways, so instead they test one thing after the other and keep running into the same proverbial brick wall at every turn; profile fluxes do not agree with constant altitude flight legs and time series look strange, etc. The text even starts with questioning the very existence of moisture inversions, which is off course fine! However, RH for the descending branch from BELUGA is not consistent with the suggested cloud outline; in the upper 50% of the cloud layer, RH < 80%; I do not believe that is the case.

Remote sensing retrieval software is wonderful and multi-sensor retrievals, like Cloudnet, has many useful features. This is, however, only true when used carefully and from an understanding of limitations and applicability. Here the authors are using Cloudnet retrievals like a very black box and it doesn't help much. So instead they bring in LES, which is perhaps an even larger black box but also doesn't help much; what is needed here is some careful thinking, experience and a new analysis strategy. The LES discussion is quite short, and I don't understand why one case is relegated to an Appendix while the other isn't, and it doesn't help at all. I would suggest to expand the LES study and make it a separate paper; base it on this study, by all means, but do the proper set of simulations to figure out the optimal configuration and then do all the different sensitivity simulations you need to extend and generalize whatever it is you find in the analysis of the observations.

There are so many ways an LES can be useful, but the way it is used here is not one of them. Multiple initial and boundary condition combinations can bring a simulation to appear similar to a single case-study profile, but there is only one that is correct and it is not always evident which one; most *appear* correct for the wrong reasons. There is much else to be said about this but most importantly, you should *never* use an LES to lend credibility to observations; it should be the other way around! If you have doubts about the observations, or how to use them, throwing in an LES does *not* help! Continuing along this line, the trajectory calculations looks intriguing, but the discussion doesn't seem to go anywhere; you need to do more to be convincing, or should just drop this line of inquiry.

What might help is to explore alternative analysis methods and/or looking at more sources of concurrent observations. I suggest looking more at the remote sensing data independently. For example, directly explore the Doppler data from the cloud radar. There are methods described in the literature how to estimate some turbulence statistics directly from the radar data (e.g. $\sigma_w$ and $\varepsilon$). An upside to this is that you can find levels where the data comes from a constant altitude for well understood portion of time; flipsides are the lack of resolution and that only one parameter can be derived. But do look at the native time resolution; not the Cloudnet-filtered data.

I would also not walk away from the slant profiles just yet, although they take really careful hands-on analysis. There are several old papers where slant profiles by aircraft have been used to tease out profiles of turbulence statistics with realistic magnitudes and shapes. It does require careful filtering, however, I submit that the vertical velocity of the platform should make aircraft profiles harder to work with than the BELUGA data. However, I would, in contrast, advise against filtering data from constant height flight legs. A numerical filter can never provide a signal with a power spectrum looking anywhere near realistic. So just give it up and use Ogive analysis instead, to analyze the magnitudes of fluxes and variances.

A word of warning, however; if the signal looks like in Figure 7a, no filtering in the world will help. The interface between the cloud and the inversion layer is like the surface of a lake and what you see here is the effect of the sensor sometimes being under and sometimes above the "surface". The resulting signal is from two different environments and filtering the signal to make it look smoother will not make those environments the same or even similar; averaging statistics for turbulence over the resulting signal is therefore meaningless, and you need to do something else.

I see no reason to expect the turbulent flux here to be in any other direction than that dictated by the gradient; counter-gradient fluxes appear in deep convective boundary layers, and this is essentially either a near neutral layer close to the upper boundary, in the cloud layer, or a stably stratified environment, in the inversion. So using the flux-gradient approach makes a lot of sense, however, I don't understand the efforts to use parameterizations of the eddy-exchange coefficient, $K_q$, based on filtered higher-order moments. Why not get it directly from the sensible heat flux and the temperature gradient? If you anyway assume that $K_q = K_H$, this should give you what you want. With the method you use, you can both measure (by eddy-covariance) and calculate (with the flux-gradient method) the sensible heat flux; if the two are different, then you can't trust the parameterized moisture flux either. However, I would say that if the gradient is positive and the flow is turbulent, there's no question in my mind the flux is negative (downward); it just stands to reason, with what we know about turbulent flows. How large it is, is a different question; one that we likely cannot get a useful answer to from one case.

Finally, many are the papers that have tried to explain peculiarities in the results with gravity waves; I have even written at least one myself and that doesn't necessarily make me proud to admit. There are, however, methods to show if what you see are indeed buoyancy waves and not just something that happens to look wavy. So – either show up or let up; either you provide some evidence that there are gravity waves present or drop that line of hand-waiving arguments all together.

---

## Referee Comment (RC2) · Anonymous Referee #2 · 27 Jul 2020

Peer review of manuscript "Case study of a humidity layer above Arctic stratocumulus using balloon-borne turbulence and radiation measurements and large eddy simulations" by Egerer et al., for potential publication in ACP.

This paper uniquely combines in situ observations, turbulence theory and associated parameterizations, and LES modeling to explore the connection between cloud top processes and overlying humidity inversions in the Arctic. The authors do a great job describing the field campaign and the methodology used to analyze the measurements and modeling studies. From a relatively limited measurement period, it is found that

[Figure]

SHIs are likely intimately connected to the cloud via turbulent kinetic energy production likely associated with cloud top radiative divergence and/or gravity wave fluctuations across the stable temperature inversion layer. These results confirm a number of observationally-based hypotheses and LES modeling studies, where it has often been speculated that SHIs play an important role in Arctic cloud physical characteristics and cloud lifetime.

I find the paper to be very well written. I am particularly delighted to see the combination of unique, state-of-the-art in situ measurements analyzed in complement with turbulence theory and LES modeling. I only have very minor concerns, which are outlined below. Once the authors address these concerns, I would be happy to support the publication of this manuscript in ACP.

General comments

1) I appreciate the discussion regarding the potential biasing of humidity inversions due to sensor wetting during the ascent through a cloud layer; this has been a caveat or concern in the community for some time, considering many of our climatological frequencies of SHI occurrences have been derived from radiosoundings from field campaigns. It is great to see the ascent/decent profiles of humidity from the BELUGA system do in fact show similar thermodynamic structures to the radio soundings.

Have any additional tests been made to attempt to isolate cases where the radiosounding-derived SHIs are potentially biased by sensor wetting, in which case these profiles could be removed from the analysis? I wonder if it would be helpful to broadly estimate the adiabatic liquid water content of the cloud layer from the thermodynamic profiles, and make a comparison with the absolute increase in specific humidity within the SHI (i.e., sensor wetting should likely not exceed the maximum LWC value in the profile). Surely the amount of sensor wetting must be limited by the maximum amount of cloud liquid water content(?).

2) The analysis and conclusions derived in this study come from really only 2 profile

cases. And even these 2 case have substantial variability in the physical properties of the inversion structures, the flux magnitude estimates, and the turbulence characteristics. I am missing an attempt by the authors to characterize or relate the flux estimates (negative) to the properties of the temperature and humidity inversion layers. How might the displacement depth between SHI base/max and level of largest infrared divergence (cooling) affect the results? I would like to see some more of this substance in the discussion Section 5.

Specific comments

Line 26: See/include reference to Devasthale et al. (2011, ACP: "Characteristics of water-vapor inversions observed over the Arctic by Atmospheric Infrared Sounder (AIRS) and radiosondes")

Line 52. The section heading "Observational" is an adjective, and therefore requires a noun to follow. Please adjust accordingly.

Line 95. It seems to me, from Fig. 2, that the other two balloon flights during the 5-7th June also correspond with the 12 UTC sounding time and have a continuous ascent and descent profile. The authors should explain, or show, why the results from this soundings and balloon profiles are not shown or described in the text. Do the profile comparisons not look as convincing as in Fig. 1?

Line 100. It would be helpful to include the cloud boundaries from Cloudnet at the time of the balloon decent as well. This may help to explain the discrepancy between RH and cloud boundaries.

Line 114-115: I am confused. I thought the Cloudnet retrievals included ceilometer base heights, MWR liquid water path estimates, and thermodynamic profiles from soundings to retrieve cloud boundaries?

Line 124-125: It would be helpful to include the cloud base and top heights (as colored symbols) on the normalized profiles, in order to show whether (and how deep) the

cloud top extended into the temperature and humidity inversion structures.

Line 145-146: Note additional studies as references: Sedlar et al. (2012, JCLIM); Shupe et al. (2013, ACP); Sedlar and Shupe (2014, ACP); Brooks et al. (2017, JGR).

Line 157: Between which depths in the layer are the Ri number calculated?

---

## Author Comment (AC1) · 9 Dec 2020

**Author's response to two anonymous reviews for ACP-2020-584**

We thank the two anonymous referees for their constructive feedback, which significantly improved the quality of the manuscript. We are well aware of how much work such a report requires. Due to the extensive modifications of the manuscript, we compose a combined author's response for both reviews. First, we summarize general revisions of the manuscript. Second, we refer to the remarks of each reviewer individually. The original reviewer comments are marked in blue color.

**General remarks**

- The structure of the manuscript has been completely revised.
- A new section about technical aspects of humidity measurements under cold and cloudy conditions has been implemented. We want to make sure that the observed humidity inversions are real and not a measurement artifact. The main reason for this was the observation of systematic humidity differences when comparing ascents and subsequent descents. We discuss error sources for the RH measurements and improve the measurements with a revised time-response correction based on further laboratory investigations.
- Due to the improved correction of humidity observations, data observed during descents are now more consistent and therefore included in the data analysis. For the descents we observed an interesting phenomenon: During all flights, the cloud base descended between ascent and descent, but in a different way. This behavior was confirmed by remote sensing. Due to the increased number of profiles with different relative locations of temperature inversion, SHI and cloud top, further scientific questions could be analyzed.
- We revised the analysis of turbulent fluxes, see specific comments below.
- We tried to focus on the novelty of our measurements rather than on uncertainties, as suggested by reviewer #1.
- Both reviewers criticize that it is hard to generalize from case studies. In our study, we document and analyze the observed cases and agree that the results should not be generalized. Further observations over a larger measurement period are needed for a more general conclusion, as stated in our summary.
- Reviewer # 1 suggested shifting the LES to a separate paper, reviewer # 2 appreciated the combination of observations and LES. We decided on a compromise and now discuss one LES case to show the impact of the SHI on the cloud, leaving potential for a more detailed separate study. See also the answer to the specific comment of reviewer # 1 below.

**Remarks to comments of referee # 1**

- Excessively long and tedious title.

We agree and change the title to "Case study of a humidity layer above Arctic stratocumulus and potential turbulent coupling with the cloud top".

- Coherent narrative instead of chronological description

We completely restructured the manuscript. We are confident that this revised version is much more narrative.

- There is no clear hypothesis to test.

We agree with this point and in the revised version, the main scientific question is raised in the introduction. We don't word it as a hypothesis, but we think this is a question of style.

- The text even starts with questioning the very existence of moisture inversions, which is off course fine! However, RH for the descending branch from BELUGA is not consistent with the suggested cloud outline; in the upper 50% of the cloud layer, RH < 80%.

We completely agree with this point, the interpretation of this profile was misleading and not convincing. In the case shown in the first version of the manuscript (old Fig. 1), the cloud extent is estimated from Cloudnet data only for the ascent. For the revised version, we use radar reflectivity raw data with a much higher temporal resolution of 3 s (30 m in vertical) (see new Fig. 2). Here it becomes clear that the balloon descended into a much lower cloud top which partly explains the low humidity in that region. However, there is a general difference in measured RH around cloud top observed during ascents and descents, which motivated us to look deeper into the data resulting in the additional chapter about humidity measurements and an improved correction algorithm.

For the new technical section about the humidity measurements, we decided to use a different day (5 June, second profile) with a constant cloud top height to clearly show the efficiency of the new corrections.

- Remote sensing retrieval software is wonderful and multi-sensor retrievals, like Cloudnet, has many useful features. This is, however, only true when used carefully and from an understanding of limitations and applicability. Here the authors are using Cloudnet retrievals like a very black box and it doesn't help much.

To a large extent, we agree with the reviewer. As a consequence, we had many discussions with our in-house experts for remote sensing observations about this topic. We agreed on using the original cloud radar data with a 30 m vertical resolution to get the most accurate estimate of cloud top development (new Fig. 1 and 2). We considered also including a comparison of remote sensing turbulence observations with BELUGA in-situ

measurements, but finally, we decided that such an analysis - although very interesting - is a different topic which we will consider in a separate manuscript. Turbulence estimated from remote-sensing is only available for in-cloud regions and we focus on the region between cloud-top and the SHI above so remote-sensing does not help very much in this context.

- So instead they bring in LES, which is perhaps an even larger black box but also doesn't help much; what is needed here is some careful thinking, experience and a new analysis strategy. The LES discussion is quite short, and I don't understand why one case is relegated to an Appendix while the other isn't, and it doesn't help at all. I would suggest to expand the LES study and make it a separate paper; base it on this study, by all means, but do the proper set of simulations to figure out the optimal configuration and then do all the different sensitivity simulations you need to extend and generalize whatever it is you find in the analysis of the observations. There are so many ways an LES can be useful, but the way it is used here is not one of them. Multiple initial and boundary condition combinations can bring a simulation to appear similar to a single case-study profile, but there is only one that is correct and it is not always evident which one; most appear correct for the wrong reasons. There is much else to be said about this but most importantly, you should never use an LES to lend credibility to observations; it should be the other way around!

We appreciate the suggestion to publish the LES study in a separate paper and will keep this option for a more detailed study on how the additional humidity is processed in the cloud layer. However, here we suggest keeping the LES discussion but shift the focus: We don't use the LES as a validation for the measurements. Instead, we focus on one LES case in order to compare with observations to show how the SHI might influence the cloudy ABL. The second LES case, as shown in the appendix, has been removed to the revised version. The technical details about the LES setup are now shifted to the appendix not to destruct the reader from the main point.

- […] the trajectory calculations looks intriguing, but the discussion doesn't seem to go anywhere; you need to do more to be convincing, or should just drop this line of inquiry.

We agree and omit the trajectory discussion at this point. It might help to explain the source for the humidity layers but this is not the focus of our paper.

- What might help is to explore alternative analysis methods and/or looking at more sources of concurrent observations. I suggest looking more at the remote sensing data independently. For example, directly explore the Doppler data from the cloud radar. There are methods described in the literature how to estimate some turbulence statistics directly from the radar data (e.g. $\sigma_w$ and $\epsilon$). An upside to this is that you can find levels where the data comes from a constant altitude for well understood portion of time; flipsides are the lack of resolution and that only one parameter can be derived. But do look at the native time resolution; not the Cloudnet-filtered data.

To broaden our expertise in remote sensing data analysis, we invited Hannes Griesche as co-author. He analyzes the PASCAL remote sensing observations. We considered including turbulence parameters, such as variance and ε derived from the cloud radar data, which gave some interesting insight into cloud dynamics (see Fig. 1 of this document). However, we realized that in-cloud turbulence does not really help to understand the coupling between the SHI and the cloud-top region due to lacking data above the cloud top. Therefore, we decided not to use this kind of analysis in this manuscript. However, these discussions about using radar data more directly helped a lot in improving the cloud top estimates from radar - see new Fig. 2.

[Figure]

Fig. 1: Eddy dissipation rate estimated from cloud radar data for the observation period.

- Flux discussion:

  - I fail to understand why knowing the size of that flux – from one case – is so important.

    We agree with the concerns about the absolute number of a single flux estimate. The discussion around these numbers and uncertainties was misleading and went in the wrong direction. We now focus more on the general shapes of the vertical profile rather than on absolute magnitudes of fluxes.

  - I would also not walk away from the slant profiles just yet, although they take really careful hands-on analysis. There are several old papers where slant profiles by aircraft have been used to tease out profiles of turbulence statistics with realistic magnitudes and shapes. It does require careful filtering, however, I submit that the vertical velocity of the platform should make aircraft profiles harder to work with than the BELUGA data.

    We stick to the slant profiles, but with slightly changed filter settings. Following arguments by Tjernström (1993) and Lenschow (1988), we set the filter window to 10 s to define the fluctuations from which the local flux is calculated. With this smaller filter window (compared to the ABL-dependent filter window of 50-100 s as applied before), we resolve the smaller structures around the SHI. The flux is then averaged over running 50 s windows on the slant profile.

○ However, I would, in contrast, advise against filtering data from constant height flight legs. A numerical filter can never provide a signal with a power spectrum looking anywhere near realistic. So just give it up and use Ogive analysis instead, to analyze the magnitudes of fluxes and variances.

Ogives are definitely an interesting tool to analyze fluxes. However, as mentioned in previous points, we focus now more on the general vertical structure of the flux profiles instead of estimating fluxes from constant level records.

○ A word of warning, however; if the signal looks like in Figure 7a, no filtering in the world will help. The interface between the cloud and the inversion layer is like the surface of a lake and what you see here is the effect of the sensor sometimes being under and sometimes above the "surface". The resulting signal is from two different environments and filtering the signal to make it look smoother will not make those environments the same or even similar; averaging statistics for turbulence over the resulting signal is therefore meaningless, and you need to do something else.

The reviewer is absolutely right, and we agree that the way the mean fluxes are estimated by filtering records as shown in Fig. 7a is fundamentally wrong. Following the argument that a single value of the flux is not meaningful in this context, we have not included another analysis technique such as Ogive analysis (although we have tried this technique). But we are convinced that a figure like the old Fig. 7 (new Fig. 13) - especially because of the remarkably constant measuring height - can give a valuable impression of the situation around the inversion, and therefore we discuss the observations based on time series. We agree that the reason for a varying $z_i$ is less important here and therefore we will refrain from a corresponding discussion at this place.

○ I see no reason to expect the turbulent flux here to be in any other direction than that dictated by the gradient; counter-gradient fluxes appear in deep convective boundary layers, and this is essentially either a near neutral layer close to the upper boundary, in the cloud layer, or a stably stratified environment, in the inversion. So using the flux-gradient approach makes a lot of sense, however, I don't understand the efforts to use parameterizations of the eddy-exchange coefficient, $K_q$, based on filtered higher-order moments. Why not get it directly from the sensible heat flux and the temperature gradient? If you anyway assume that $K_q = K_H$, this should give you what you want. With the method you use, you can both measure (by eddy-covariance) and calculate (with the flux-gradient method) the sensible heat flux; if the two are different, then you can't trust the parameterized moisture flux either. However, I would say that if the gradient is positive and the flow is turbulent, there's no question in my mind the flux is negative (downward); it just stands to reason, with what we know about turbulent flows. How large it is, is a different question; one that we likely cannot get a useful answer to from one case.

We adopt the reviewer's suggestion and calculate now $K_H$ from the slant profile measurements. We had some internal discussion if it is worth to calculate $K = K(z)$ or to estimate a single $K$ for the region of main interest. With a constant $K$, we definitively underestimate the flux in the more turbulent cloud layer, but in that region we would have to apply some careful averaging to smooth the local gradients avoiding too much scatter for the $K$ values. We, therefore, decided to estimate $K$ just around the base of the SHI and use this value for the entire profile. These $K$ values differ only slightly among the different days, which gives us some confidence that the method is robust.

- Finally, many are the papers that have tried to explain peculiarities in the results with gravity waves; .... There are, however, methods to show if what you see are indeed buoyancy waves and not just something that happens to look wavy. So – either show up or let up; either you provide some evidence that there are gravity waves present or drop that line of hand-waiving arguments all together.

We deleted the discussion of possible gravity waves and instead followed the reviewer's argumentation that $z_i$ moves up and down around the instrument, producing those temperature variations. We agree that for our manuscript the exact reason for the variability of $z_i$ is of less importance.

**Remarks to comments of referee # 2**

- 1) I appreciate the discussion regarding the potential biasing of humidity inversions due to sensor wetting during the ascent through a cloud layer; this has been a caveat or concern in the community for some time, considering many of our climatological frequencies of SHI occurrences have been derived from radiosoundings from field campaigns. It is great to see the ascent/decent profiles of humidity from the BELUGA system do in fact show similar thermodynamic structures to the radio soundings. Have any additional tests been made to attempt to isolate cases where the radiosounding-derived SHIs are potentially biased by sensor wetting, in which case these profiles could be removed from the analysis? I wonder if it would be helpful to broadly estimate the adiabatic liquid water content of the cloud layer from the thermodynamic profiles, and make a comparison with the absolute increase in specific humidity within the SHI (i.e., sensor wetting should likely not exceed the maximum LWC value in the profile). Surely the amount of sensor wetting must be limited by the maximum amount of cloud liquid water content(?).

We think that the sensor wetting can exceed the maximum LWC in the cloud, as liquid water can accumulate on the sensors. Hence, it is difficult to quantify wet-bulbing, as we don't have an indicator for the extent of wetting. We could not identify wet-bulbing events in the radiosoundings we analyzed. Instead, one case in our BELUGA measurements, where wet-bulbing probably occurred, is the 12 June (see Fig. 2 of this document, not included in the manuscript), where RH increased by almost 10% at cloud top. On this day, we observed wet sensors when they returned to the ground. However, this RH increase causes only a small increase in $q$ as part of the actual SHI.

[Figure]

Fig. 2: Vertical profiles for 12 June 2017 (not part of the manuscript).

- 2) The analysis and conclusions derived in this study come from really only 2 profile cases. And even these 2 case have substantial variability in the physical properties of the inversion structures, the flux magnitude estimates, and the turbulence characteristics. I am missing an attempt by the authors to characterize or relate the flux estimates(negative) to the properties of the temperature and humidity inversion layers. How might the displacement depth between SHI base/max and level of largest infrared divergence (cooling) affect the results? I would like to see some more of this substance in the discussion Section 5.

We address this comment in the new sections 4 and 5 by discussing the descents, where the SHI relates differently to temperature inversion height and cloud top,. However, we cannot relate a flux magnitude to the SHI properties, as we focus on the vertical structure of fluxes rather than a number for cloud-top fluxes.

- Line 26: See/include reference to Devasthale et al. (2011, ACP: "Characteristics of water-vapor inversions observed over the Arctic by Atmospheric Infrared Sounder(AIRS) and radiosondes")

Thank you for the reference to this paper about SHIs from radiosondes and satellite data under clear-sky conditions. We inserted the reference in the introduction.

- Line 52. The section heading "Observational" is an adjective, and therefore requires a noun to follow. Please adjust accordingly.

We changed the heading to "Observations".

- Line 95. It seems to me, from Fig. 2, that the other two balloon flights during the 5-7th June also correspond with the 12 UTC sounding time and have a continuous ascent and descent profile. The authors should explain, or show, why the results from this soundings and balloon profiles are not shown or described in the text. Do the profile comparisons not look as convincing as in Fig. 1?

We included the new Fig. 2 to show the single flight profiles more in detail and with regard to the cloud. The first flights of 6 and 7 June have constant height steps on the descent. For a comparison between ascent and descent, we now show the second, smaller but continuous profile of 5 June with a constant cloud top height. However, we also included the radiosoundings in the vertical profiles of mean parameters for each day.

- Line 100. It would be helpful to include the cloud boundaries from Cloudnet at the time of the balloon decent as well. This may help to explain the discrepancy between RH and cloud boundaries.

We now discuss the cloud tops (based on irradiance data) on the ascents and descents for all flights in detail in the new Sect. 4. To discuss the humidity measurements (with a comparison of ascent and descent), we now show another day (5 June) with constant cloud top height.

- Line 114-115: I am confused. I thought the Cloudnet retrievals included ceilometer base heights, MWR liquid water path estimates, and thermodynamic profiles from soundings to retrieve cloud boundaries?

In the first manuscript version, we showed the cloud base from a separate Ceilometer, which was part of the Polarstern standard meteorological observations. We now use the cloud base data derived from the lidar PollyXT near-field channel, which is part of the Cloudnet sensor suite. The lidar has a resolution of 7.5 m and 30 s. Using native lidar data, not processed with the Cloudnet algorithm, allows detecting cloud base heights below the lowest Cloudnet range gate of 155 m, which is determined by the cloud radar.

- Line 124-125: It would be helpful to include the cloud base and top heights (as colored symbols) on the normalized profiles, in order to show whether (and how deep) the cloud top extended into the temperature and humidity inversion structures.

We included a separate panel to show the cloud top height (derived from irradiance profiles). We observe almost no cloud tops extending into the inversion.

- Line 145-146: Note additional studies as references: Sedlar et al. (2012, JCLIM);Shupe et al. (2013, ACP); Sedlar and Shupe (2014, ACP); Brooks et al. (2017, JGR).

We included the suggested references. However, we did not observe that the cloud tops penetrated into the SHIs, as discussed in those studies.

- Line 157: Between which depths in the layer are the Ri number calculated?

This question is answered by the new columns in Fig. 8-10, showing the vertical profile of the Richardson number.

---

## Author Response (AR2)

**Author's response to one anonymous review for ACP-2020-584, second revision**

Reviewer # 1

We would like to thank the reviewer once again for his/ her helpful comments, which have significantly improved the quality of the revised version. We hope that we were able to address all critical points in the manuscript. Below you will find our responses to the individual points (the reviewer's comments are highlighted in blue).

Three overriding concerns:

- The main message: When reading the manuscript, the narrative shifts from the effects of SHI, to questioning the very existence of SHI and then back to the effects again. The introduction covers the SHI in previous studies and sets the stage for the analysis. Then with the long section with the discussion of potential measurement errors the very existence of SHI is called into question and it is concluded they do exist. This is a relief given the intro; there wouldn't be much of a paper if the conclusion had been the opposite. However, the discussion of potential errors is then mostly forgotten in the rest of the text, which reverts back to accepting their existence; it is not even mentioned in the concluding section. This makes the section with the error analysis (Section 3) seem like a long and embedded appendix; it doesn't fit well into the rest of the text. There's nothing fundamentally wrong with the text seen section by section; I'm just sensitive to the narrative when looking at all the sections as part of one paper.

Thank you very much for bringing this point to our attention. We agree with this concern and in the new version, we tried to better integrate section 3 into the scope of the manuscript by mentioning the issue of possible measurement errors already in the introduction and incorporating the technical part also in the summary and abstract. We also swapped sections 2.2 (BELUGA) and 2.3 (Observation period) to bring the technical sections closer together.

- The error analysis (Section 3): This is worthwhile but very detailed description also with some rather trivial content. Does it have to be this long(?); it's a third of the main text. With this degree of detail, it could have been published separately as a technical note. The authors list three potential errors. The solar heating is not considered further, leaving the wet-bulb:ing and time constant. Then, if I read this correctly, the wet-bulbing seems to be folded into a time-constant problem; then at the end it is not again (see final sentence). But viewing wetting this way makes things so much more complicated; it means that the system has two time constants; one related to the time it takes for the wetting to evaporate and one relating to the time constant of the instrument. In fact, there may even be three time constants; these two plus one related to the instrument housing. I recommend that either Section 3 is revisited and rewritten so that it fits the purpose of this paper, or that it

is extracted from this paper and published separately. If it is kept as a part of this manuscript, the results should be reflected in the following text and especially in the summary section.

Thank you again for this point. We agree that section 3 is quite long compared to the more science-related sections. However, we think that the topic should be presented together with this study and would like to keep the content, but modify it. We shortened the section and left out some less important content, such as the figure showing the influence of the time constants on $q$. We shifted the subsection about determining the time constants of our humidity sensor in the appendix, as it is additional information and no prerequisite for the rest of the paper. The aim of this section is to show what can distort the SHI observations and to make sure that the SHI is not the result of one of these errors. We show that we can quantify and eliminate the time lag errors and minimize the influence of solar heating by using measurements inside the same sensor housing. We cannot quantify wet-bulbing and rule out that this effect exists in our observations, but we can exclude it as the main reason for the observed SHIs because the SHIs are seen also during the descents. This is an advantage of BELUGA measurements compared to radiosoundings. We try to better transport this message in the revised text.

- The LES study: I indicated previously that I didn't think the LES study fitted in this manuscript. The way the revised paper comes through, I take that back. While I do feel that one can always get an LES to agree acceptably with observations if one tries hard enough (there are so many degrees of freedom to play with and usually not enough observations to constrain) the trick is to use the LES for something valuable and with the with and without SHI comparison I feel the authors succeed with only these two runs. They may both be way off to reality in some aspect, but they are then off the same way! But more work is still needed to make it fit in the paper. The text introduces just about enough information about the LES to irritates me to have to go to the Appendix to get the rest. I get to know the size of the domain, but not the resolution; for that I need to go to the Appendix. I can't find information on how the LES is initialized and get no useful information on what the authors mean by "Lagrangian"; yet this particular information is repeated in both the text and the Appendix. The Appendix say the LES is "constrained by the soundings", but those were done at the location of Polarstern which is the trajectory end point. So how is that compatible with a Lagrangian perspective? In short, the text describing the LES and the experimental setup needs to be revisited; I would either put all technical information in the Appendix or expand the Section in the text to get rid of the Appendix.

We are glad that the reviewer shares our opinion that including an analysis of two targeted LES experiments, one with and one without an SHI, is meaningful and brings added value to this study. In the revision, we tried to address all of the reviewer's comments concerning the discussion of the LES. Firstly, as recommended, the technical description of the numerical experiments has now completely moved to the Appendix; the details mentioned there allow complete and independent reconstruction of the conducted experiments. This includes information about the Lagrangian setup (which is well established and has often been used in idealized LES studies of cloudy boundary layers), the model initialization, and the adjustments made to yield good agreement with the observed basic state and structure of the cloudy mixed layer. The main body of the text now only includes a

general introduction to the simulations, as well as a brief motivation for their use in this study. We hope that we managed to strike the right balance in this respect and that this has helped in improving the flow of the narrative.

- The SHI gap: All the results from the LES and the observations concerning the case where the SHI is disconnected from the cloud top makes sense to me, but the whole thing begs the question: Why? With the accepted hypothesis on the existence of SHIs being related to large scale advection of a deeper and moister upstream PBL that adjusts to the shallower PBL forcing over the sea ice, I don't immediately quite see how this could happen. Where did the moisture go? In to the cloud and precipitated out, while the cloud top then proceeded to evaporate?

We agree with the reviewer's opinion that this observation raises many questions and also leaves most of them open. In the end, the structure of the SHI remains essentially unaffected, while the cloud top temporarily - see the radar data - drops sharply, leaving the gap behind. We have tried to find some indication of this boundary layer behavior in the continuous ground data but without success. We, therefore, decided to formulate this observation as an open question for further investigation. Unfortunately, this is not satisfying for us either, but together with our radar colleagues, we will continue to pursue the question of what could be the cause for such a sudden drop of the cloud top and whether this has been observed in this form more often.

Minor comments:
- Line 10 and elsewhere: There is considerable discussion of "latent heat flux", but isn't it the turbulent flux of water vapor that is important to this paper. Not the effects on or by the heat transport (energy) but the transport of mass; water vapor. So why convert it to W m-2?

We agree that the mass flux of water vapor is of interest. Since it is proportional to the latent heat flux (by $L_v$), we use the latent heat flux in W/m$^2$ for consistency with the other flux values.

- Line 35: Why confine it to advection from continents? It could equally well be marine air from south of the ice margin.

Agreed, we removed "continental".

- Line 43: "Despite their importance …" implies an causality between knowledge and importance that isn't necessary there. Some of the most important issues in science have turned out to be the most difficult to solve. Take "climate sensitivity" as an example.

That's probably true and we agree to remove the first part of the sentence.

- Line 79 and elsewhere: The sonic anemometer provides a so called "sonic temperature". This is not equivalent to the virtual temperature, although it is close enough, especially in dry environments (not necessarily low RH but low q).

Thanks for the hint, we added in the manuscript: "Especially at low specific humidity, the sonic temperature is close to the virtual temperature, which will be used in the following.".

- Lines 79-80: Considering what comes in Section 3, this is not nearly enough discussion of these sensors. It wasn't until the end of Section 3 I realized the housing of the T/RH sensors may be a problem.
  Lines 219-224: See above; it's not until here that I get the information that there might be a problem with the instrument housing.

We added some information about the RH sensor and its housing combined with internal temperature measurements in section 2.2.

- Lines 181-187: This paragraph is actually a repetition of the previous discussion. Setting the time constant of one sensor to zero, which is already done, is consistent with setting it to any other value much shorter than the other. And setting both to zero is – in a relative sense – the same as setting both to any other single value, say for example 60 s.

The paragraph was shortened to provide only new and essential information. The figure showing the sensitivity to different time constants was removed, to omit redundant information.

- Lines 213-214: How can the warming lead to a change in RH when RH is the measured variable?

Thanks for the comment, this was misleading. We changed the sentence to "Furthermore, the sensor underestimates RH in the cloud on the descent, which might indicate solar heating.".

- Line 224: Confusing; reading the preceding text, I though wet-bulb:ing was THE problem, and that is what was considered above?

We agree the formulation was not clear enough. The revised text reads: "We can exclude wet-bulbing as the main reason for the observed SHIs because the SHI is also present during the descent. The influence of solar heating and time-lag errors is minimized. Our conclusion also strengthens the confidence in SHIs as frequently observed by radiosondes."

- Figure 7: Please display the cloud base also in Figure 7c

Good point, the figure now shows also the cloud base.

- Line 237: I disagree; while near-neutral through the PBL, it is weakly stable through the whole layer, and there is no easily distinguishable point where this increases below the inversion base.
  Lines 236-239: Note that the scaling used here does not apply to the lower free troposphere, so there is no reason a priori that the profiles above the PBL top should be similar or comparable.

We re-phrased the paragraph with the suggested points.

- Line 244: Disagree again; there are clear capping temperature inversions in all the profiles. Some profiles may have embedded internal structure but that is not the same as not showing a "clear temperature inversion".

We removed the formulation of "no clear temperature inversion".

- Figures 8-10: Why is there sometimes such a large absolute difference between the sounding and the Beluga temperature profiles? Sometimes the sounding is several degrees colder that the tethered sounding; this presumably also affects the specific humidity profiles.

A comparison to ground data (mast, ship) showed no systematic difference to BELUGA near-surface values. The temperature difference seems to be greater at higher altitudes. We explain the temperature difference by ABL variability. The strongest difference between radiosonde and balloon is observed on 5 June: Here some short but strong warming events occur (which are visible in mast data).

- Lines 280-281: A difference of 20 m could well be just coincidence; the cloud top is not at one fixed height but rather goes up and down following the characteristics of the up- and downward motions of the turbulent eddies.

We added: "... 20m below $z_i$, which could possibly result from cloud top heterogeneity.".

- Section 5.3: This Section doesn't really add much information other than as a motivation for using slant profiles instead. Therefore, either move it up as Section 5.1 and use it that way. Or drop it…

We changed the order of the sections and now use the old section 5.3 as new section 5.1 to motivate the vertical profile measurements.

- Lines 387-388: The conclusions on the distance to the PBL top rests on an assumption on a constant vertical gradient. I submit that the sections of the time series may be equally semi-constantly distant to the ABL top, but as the latter is slowly descending, the fluctuations change character from going in and out of the PBL to being entirely inside the inversion.

We took up this suggestion and changed the text accordingly.

- Lines 413-414: The simulated dq = 0.6 g kg^-1 is a factor of two smaller than the observed ; this is hardly "close to".

The SHI strength was adapted to the radiosonde observation of dq = 0.9 g kg^-1, which is closer than the BELUGA observation. We changed that in the manuscript.